# Engineering the Marine *Pseudoalteromonas haloplanktis* TAC125 via the pMEGA Plasmid Targeted Curing Using PTasRNA Technology

**DOI:** 10.3390/microorganisms13020324

**Published:** 2025-02-02

**Authors:** Angelica Severino, Concetta Lauro, Marzia Calvanese, Christopher Riccardi, Andrea Colarusso, Marco Fondi, Ermenegilda Parrilli, Maria Luisa Tutino

**Affiliations:** 1Dipartimento di Scienze Chimiche, Complesso Universitario Monte S. Angelo, Università degli Studi di Napoli Federico II, Via Cintia 4, 80126 Napoli, Italy; angelica.severino@unina.it (A.S.); marzia.calvanese@unina.it (M.C.); ermenegilda.parrilli@unina.it (E.P.); 2Dipartimento di Biologia, Università degli Studi di Firenze, Via Madonna del Piano 6, 50019 Firenze, Italy; christopher.riccardi@unifi.it (C.R.); marco.fondi@unifi.it (M.F.); 3Department of Integrative Structural and Computational Biology, The Scripps Research Institute, San Diego, CA 92037, USA; acolarusso@scripps.edu; 4Istituto Nazionale Biostrutture e Biosistemi I.N.B.B., Via dei Carpegna, 19, 00165 Roma, Italy

**Keywords:** *Pseudoalteromonas haloplanktis* TAC125, cold-adapted bacteria, pMEGA, megaplasmid, strain engineering, plasmid curing, PTasRNA gene silencing

## Abstract

Marine bacteria that have adapted to thrive in extreme environments, such as *Pseudoalteromonas haloplanktis* TAC125 (*Ph*TAC125), offer a unique biotechnological potential. The discovery of an endogenous megaplasmid (pMEGA) raises questions about its metabolic impact and functional role in that strain. This study aimed at streamlining the host genetic background by curing *Ph*TAC125 of the pMEGA plasmid using a sequential genetic approach. We combined homologous recombination by exploiting a suicide vector, with the PTasRNA gene-silencing technology interfering with pMEGA replication machinery. This approach led to the construction of the novel *Ph*TAC125 KrPL^2^ strain, cured of the pMEGA plasmid, which exhibited no significant differences in growth behavior, though showcasing enhanced resistance to oxidative stress and a reduced capacity for biofilm formation. These findings represent a significant achievement in developing our understanding of the role of the pMEGA plasmid and the biotechnological applications of *Ph*TAC125 in recombinant protein production. This opens up the possibility of exploiting valuable pMEGA genetic elements and further advancing the genetic tools for *Ph*TAC125.

## 1. Introduction

Much of life on Earth has evolved to thrive in cold environments, which constitute the most widespread habitat on the surface of our planet [1]. Despite the inherent challenges associated with surviving in these cold areas, psychrophilic (or psychrotolerant, or cold-adapted) microorganisms—including bacteria, archaea, yeasts, unicellular algae, and fungi—have remarkably adapted to life in these harsh conditions [2]. These microorganisms can survive at temperatures below 7 °C, with optimal growth typically occurring at higher temperatures ranging from 4 to 20 °C [3].

Studies of bacterial communities in such environments have shown the widespread distribution and abundance of *Gamma-proteobacteria*, which predominantly colonize both deep-sea and surface waters, as well as sea–coastal sediments [4]. Among *Gamma-proteobacteria*, the genus *Pseudoalteromonas* is particularly noteworthy because of its extensive distribution in marine environments, where it represents a significant portion of the ocean bacterial community (2–3% in the surface ocean and 14% in the deep sea) [5]. Additionally, members of the *Pseudoalteromonas* genus are renowned for their exceptional environmental adaptability, enabling them to thrive in these extreme habitats [6,7].

Psychrophilic microorganisms counteract the deleterious effects of low temperatures by developing specific strategies in the form of finely tuned structural changes at the level of, for example, their membranes, constitutive proteins, and enzymes, which explain their broad environmental adaptability [2,7]. In this context, *Pseudoalteromonas haloplanktis* TAC125 (*Ph*TAC125) is amongst the most valuable marine microorganisms, and it is considered a model organism of cold-adapted bacteria, reaching a high cellular density at the optimal temperature of 15 °C [8,9]. The fully sequenced and annotated multipartite genome of *Ph*TAC125 (composed of two chromosomes and pMtBL, a cryptic plasmid) [6,10] has paved the way for understanding cold adaptation strategies in this bug, which in turn has proved to be (a) a source of valuable bioactive compounds, such as anti-biofilm molecules [11,12] and antimicrobials [13,14], (b) a tool for bioremediation [15,16], and (c) an exceptional host for recombinant production of “difficult proteins” preventing insoluble protein aggregates [17,18,19,20]. In this last regard, several expression vectors, based on pMtBL replication signals and with either constitutive or inducible promoters [21,22], were developed to enable the recombinant production of many proteins, both in planktonic [9,21] and biofilm cultures [23]. Additionally, different genetic tools were designed for *Ph*TAC125, including (a) a strategy for the construction of insertion/deletion genomic mutants exploiting homologous recombination, in which Giuliani and coworkers elaborated a technique for allelic exchange and/or gene inactivation by in-frame deletion and the use of a counterselection marker harnessing a non-replicative plasmid [24], and (b) a conditional gene silencing system by PTasRNA technology developed by Lauro et al. to effectively down-regulate chromosomal gene expression of *Ph*TAC125 and achieve high levels of gene silencing [25].

Interestingly, in 2019, the resequencing of the *Ph*TAC125 genome using third-generation sequencing technologies uncovered a novel large plasmid, pMEGA, which is 64,758 bp in size and contains 52 open reading frames (ORFs) [10]. pMEGA shows limited similarity to *P. haloplanktis* chromosomes, but it shares a substantial nucleotide similarity with plasmids found in other marine bacteria, including *Pseudoalteromonas arctica* and *Pseudoalteromonas nigrifaciens*, suggesting a unique evolutionary origin and potential adaptation to cold marine environments [10]. pMEGA is a low-copy-number plasmid and is considered a non-conjugative one. It encodes essential replication and stability functions, such as the RepB replication initiator protein, and the ParA and ParB proteins involved in partitioning and stability. Qi et al. leveraged the high nucleotide sequence conservation between the pMEGA plasmid and those from *P. arctica* and *P. nigrifaciens* to highlight similarities in the replication and partitioning functions, hypothesizing a rolling-circle replication (RCR) mechanism with a common regulation system [10]. pMEGA encodes two type II toxin–antitoxin systems, the HipBA and the hybrid YefM-ParE [26]; it harbors genes coding for defense mechanisms against bacteriophages, including type I and type IV restriction–modification systems as well as several proteins with a role in cell metabolism (TonB-dependent receptors, an aminotransferase, a nitronate monooxygenase, an epimerase, and an acetyltransferase).

More recently, pangenome studies revealed that multipartite genomes are widely distributed in the class *Gamma-proteobacteria*, in which the major representatives are *Vibrionaceae* [27] and *Pseudoalteromadaceae* (prevalently *Pseudoalteromonas* spp.) [28]. Using a phylogenetic approach and timescale analysis of *Pseudoalteromonas*, Liao et al. (2019) showed that chromosomes and chromids (carrying essential housekeeping genes localized on the chromosomes in other species) have always coexisted and that chromids originated from a megaplasmid that, over time, obtained essential genes, enhancing the hypothesis that megaplasmids are commonly distributed in this genus [6,29]. However, some megaplasmids, like pMEGA, that lack their own conjugative machinery either rely on the conjugative systems of other plasmids for transmission (i.e., they are mobilizable) or may not be mobile at all. In the long run, these plasmids may either impose a metabolic burden and evolve into efficient selfish mobile elements or become mutualists, dependent on the host for replication and transmission [30]. Loss or removal of the megaplasmid might help highlight other aspects of the physiological and functional roles of that conserved megaplasmid in *Ph*TAC125.

In this study, we explored the genomic and transcriptomic landscape of the pMEGA plasmid to build a finely tuned strategy tailored to its curing. We developed a sequential genetic scheme through the combination of two different genetic methodologies previously developed in *Ph*TAC125. We first edited pMEGA by integrating the pAT suicide vector carrying a selection marker, necessary to distinguish wild-type from cured bacteria. Then, we used PTasRNA gene-silencing technology to target and interfere with the replication machinery of the plasmid. We successfully removed the pMEGA plasmid from the bacterium, which resulted in the construction of a novel strain named KrPL^2^ (Figure 1). The phenotypical characterization showed that this strain exhibited comparable growth dynamics to its progenitor strain, demonstrating that the removal of pMEGA did not significantly impact the strain growth or survival across different temperatures and media conditions. Additionally, plasmids that share similarities in their replication or partitioning systems cannot stably coexist in the same bacterial cell, leading to plasmid incompatibility [31]. Hence, further evaluations of plasmid replicability, stability, and copy number revealed that, despite the removal of pMEGA, the inability of *rep*-based plasmids to replicate persisted, indicating the involvement of additional mechanisms beyond replication-associated genes. This study also confirmed that the removal of pMEGA did not affect the pMtBL-derived plasmid stability or copy number under the conditions tested. Also, this work demonstrates the efficacy of PTasRNA technology in curing plasmids with rolling-circle replication, extending the application of this methodology for future applications in genetic manipulation and functional studies in psychrophilic bacteria.

## 2. Materials and Methods

### 2.1. Bacterial Strains, Media, and Plasmids

*Escherichia coli* TOP10 (mcrA, Δ(mrr-hsdRMS-mcrBC), ϕ80lacZ (del) M15, ΔlacX74, deoR, recA1, araD139, Δ(ara-leu)—7697, galU, galK, rpsL (SmR), endA1, nupG) was used as a host for cloning procedures. *E. coli* strain S17-1(λpir) [thi, pro, hsd (r− m+) recA::RP4-2TcR::Mu KmR::Tn7 TpR SmR λpir] [21,32] was used as a donor in intergenic conjugation experiments. *Pseudoalteromoans haloplanktis* TAC125 (*Ph*TAC125) KrPL, a strain cured from the endogenous pMtBL plasmid [21], was used as a host for the homologous recombination experiment. The *Ph*TAC125 KrPL insDNApolV mutant was used as a host for the expression of the antisense RNA in the conditional gene-silencing protocol. *Psychrobacter* sp. TAD1 white (NCBI:txid81861) was used as a control of the incompatibility assay already available in the laboratory.

*E. coli* strains were grown in LB broth (10 g/L bacto-tryptone, 5 g/L yeast extract, 10 g/L NaCl) at 37 °C, and the recombinant strains were treated with either 34 μg/mL chloramphenicol or 100 μg/mL ampicillin, depending on the selection marker of the vector. *Ph*TAC125 strains were cultured in TYP medium (16 g/L bacto-tryptone, 16 g/L yeast extract, 10 g/L NaCl), or GG 10-10 medium (10 g/L L-glutamic acid monosodium salt monohydrate, 10 g/L D-gluconic acid sodium salt, 10 g/L NaCl, 1 g/L NH_4_NO_3_) [9] for the recombinant production of PTasRNA*repB* and GG 5-5 (5 g/L D-gluconate, 5 g/L L-glutamate, 10 g/L NaCl, 1 g/L NH_4_NO_3_), according to the experimental setup. During the experiments, GG 10-10 and GG 5-5 were supplemented with Schatz salts (1 g/L K_2_HPO_4_, 200 mg/L MgSO_4_·7H_2_O, 5 mg/L FeSO_4_·7H_2_O, 5 mg/L CaCl_2_) in sterile conditions.

When necessary, chloramphenicol was added to solid and liquid media at 12.5 μg/mL and 25 μg/mL concentrations, respectively. Ampicillin was always used with a concentration of 100 μg/mL.

Competent cells of *E. coli* strains were produced using the calcium chloride method and transformed with the appropriate vector through the heat-shock transformation method [33]. *Ph*TAC125 cells were transformed with the appropriate vector via intergeneric conjugation, as reported in Tutino et al. (2001) [32].

The recombinant plasmids used in this work are listed in Table 1. pMAV was used during the plasmid copy number assessment, and the construction is reported in Sannino et al., 2017 [9]. pKT240 and pUC-oriT-pTAUp plasmids along with pMAV plasmid were employed during the incompatibility assay. pAT-*eGFP* was used for the construction of pAT-VS-HR*umuC,* which was designed for the pMEGA curing strategy. pB40-79-PTasRNA*lon*, pB40-79C-PTasRNA*lon*, and pB40-79-PTasRNA*repB* were designed and used in the PT-antisense interference experiment for the selection of the pMEGA cured clones. pB40-79-PTasRNA*rep*B was also used for the heterologous plasmid stability assessment.

### 2.2. Bioinformatic Analyses and Transcriptomic Data Collection

To identify regions of similarity between pMEGA and other prokaryotic genomes, an initial BlastN similarity search against the NCBI nr/nt nucleotide database was performed using the National Institutes of Health (NIH) web interface in its default parameters [35]. Similarly, BlastP similarity searches of the *Ph*TAC125 proteome were then used to identify pMEGA proteins homologous to *Pseudoalteromonas* sp. KG3 and *Pseudoalteromonas* sp. PS1M3 through the National Center for Biotechnology Information (NCBI) non-redundant protein sequence (nr) database.

For the PTasRNA interference assay, prediction of RNA secondary structures was performed using the mFold website with default settings [36]. Gene expression data for *Ph*TAC125 cultured in two temperature conditions (0 °C and 15 °C, respectively) were collected from Riccardi et al. [37], and the information relative to the pMEGA transcriptome was kept for further analyses. Sequence annotations for the pMEGA coding sequence (CDS) and FASTA were downloaded from the NCBI, using accession number NZ_MN400773.1, last accessed on 10 October 2024.

### 2.3. Plasmid Construction

The suicide vector pAT-VS-HR*umuC* was obtained through gateway cloning procedures [38]. The HR*umuC* homology region of 259 bp located on the pMEGA plasmid was PCR amplified (Phusion High-Fidelity DNA polymerase—Thermo Fischer Scientific, Waltham, MA, USA) using the bacterial genomic sample of *Ph*TAC125 as the DNA template, which includes the pMEGA plasmid, and a pair of forward and reverse oligonucleotides listed in Appendix A (*umuC*_*Not*I fw, *umuC*_*Asc*I rv) (Eurofins Genomics, Ebersberg, Germany). The PCR fragment was purified, subjected to *Not*I/*Asc*I double digestion, and cloned inside the pAT-*eGFP* vector previously hydrolyzed with the same restriction enzymes. The homology region sequence was inserted between *oriC* and *oriT* of the pAT-*eGFP* vector, resulting in a plasmid lacking the psychrophilic origin of replication, thus yielding the pAT-VS-HR*umuC* suicide vector.

The pB40-79-PTasRNA*lon* vector*,* a derivative of pB40-79C-PTasRNA*lon* used in our previous work [25], was constructed through a cut-and-paste procedure where both plasmids were double digested with *SphI*/*AscI* restriction enzymes. These cleavages generate a fragment A of 2276 bp carrying the *ori*C and *amp*R regions, whereas fragment B of 2815 bp delivers the paired termini and antisense region, along with the *oriR*, *oriT*, multi cloning site (MCS), and *Ph*LacR and *placZ* promoter regions. Joining the two fragments led to the construction of the pB40-79-PTasRNA-*lon* vector, in which the chloramphenicol selection marker was replaced with the ampicillin resistance gene.

The construction of pB40-79-PTasRNA*rep*B was carried out by exploiting *Pst*I and *Xho*I restriction sites in pB40-79-PTasRNA*lon*. These sites were purposefully designed by Lauro et al. to directly exchange antisense sequences without altering the paired termini sequences upstream and downstream of the antisense RNA. asRNA*rep*B was PCR amplified (Phusion High-Fidelity DNA polymerase—Thermo Fisher Scientific, Waltham, MA, USA) using the pMEGA plasmid as a template and a pair of forward and reverse oligonucleotides listed in Appendix A (*Xho*I-asRNA*rep*B fw, *Pst*I-asRNA*rep*B rv) (Eurofins Genomics, Ebersberg). The PCR fragments and pB40-79-*Phlon* were double hydrolyzed with *Pst*I/*Xho*I restriction enzymes, resulting in the pB40-79-PTasRNA*rep*B vector. The nucleotide sequence of the synthetic asRNA*repB* used in this work is listed in Appendix A. A bacterial DNA kit (D3350-02, E.Z.N.A™, OMEGA bio-tek, Norcross, GA, USA) was used for DNA genomic extraction, following the manufacturer’s instructions.

### 2.4. PTasRNA Interference

Gene silencing using PTasRNA*rep*B was performed by culturing the bacteria in 20 mL GG 10-10 medium, as described in Section 2.1, supplemented with 100 μg/mL ampicillin in 100 mL Erlenmeyer flasks. Also, the non-recombinant Krpl *ins*DNApolV strain was used as a negative control for the experiment, grown in non-selective conditions.

Strains were streaked on TYP-agar plates supplemented with 100 μg/mL ampicillin and incubated at 15 °C for three days. A single colony was inoculated in 3 mL TYP medium supplemented with 100 μg/mL ampicillin at 15 °C overnight in a 25 mL inoculum glass tube. Cultures were diluted to 1:100 in 10 mL GG 10-10 supplemented with Schatz salts and 100 μg/mL ampicillin in a 100 mL Erlenmeyer flask and incubated at 15 °C with shaking. Following that, 0.35 OD_600_/mL dilution in 20 mL GG 10-10 medium supplemented with Schatz salts and 100 μg/mL ampicillin was performed, and the diluted product was incubated at 15 °C for 8 h in a 100 mL Erlenmeyer flask with shaking.

Inoculum was produced at 0.1 OD/mL, and induction was carried out with 10 mM isopropyl β-d-1-thiogalactopyranoside (IPTG) when the cells reached a value of 1.5 OD/mL, as reported by Lauro et al. [25]. Samples were collected at 8, 24 and 32 h post-induction. We diluted 0.1 OD/mL (1–2 × 10^7^ CFU/mL) cultures at 1:10^4^, and 100 μL was seeded on non-selective TYP agar. Incubation was performed at 15 °C for 3 days, obtaining about 100–200 colonies. The colonies grown from each plate (8, 24, and 32 h post-induction) were replicated on a 70 mL TYP-agar plate in the presence and absence of chloramphenicol as a selective agent and then incubated at 15 °C.

The clones that did not replicate on TYP agar supplemented with chloramphenicol but easily grew on TYP agar without the selective agent produced PTasRNA*repB*. These clones were subjected to PCR colony screening. A single colony of *Ph*TAC125 was resuspended in 30 µL of deionized MilliQ water, boiled for 10 min, and spun down for 10 min. The supernatant (1 µL) was used as a DNA template for the PCR reaction using Taq Polymerase (Thermo Scientific, Waltham, MA, USA) All the primers and fragments used are listed in Appendix A (prom7 fw, prom 7 rv, CDS49 fw, CDS49 rv, 5′*umuC* fw, *umuC*_*Asc*I rv).

### 2.5. Growth Culture Conditions at 0 °C, 15 °C, and 20 °C

Each strain was cultured at 0 °C, 15 °C, and 20 °C temperatures in TYP medium and GG 5-5 media supplemented with Schatz salts. The strains were streaked on a TYP-agar plate and incubated at 15 °C for three days.

A single colony was inoculated in 3 mL TYP medium at 15 °C overnight in a 25 mL inoculum glass tube. Cultures at 15 °C were performed by diluting in 10 mL TYP or GG 10-10 supplemented with Schatz salts in a 100 mL Erlenmeyer flask, and the product was incubated at 15 °C with shaking. Following that, a 0.35 OD/mL dilution in 20 mL TYP medium or GG 5-5 medium supplemented with Schatz salts was performed, and the culture was incubated at 15 °C for 8 h in a 100 mL Erlenmeyer flask with shaking. The inoculum was produced at 0.1 OD/mL and 15 °C in 50 mL TYP or GG 5-5 medium supplemented with Schatz salts in a 250 mL Erlenmeyer flask with shaking. Points of the growth curves were collected every 3 h.

Cultures at 20 °C and 0 °C were carried out by pre-adapting the pre-inoculum at the desired temperature, to avoid drastic changes in growth conditions. At 20 °C, a single colony was inoculated in 3 mL TYP, and this was incubated at 15 °C overnight. Following that, the culture was diluted 0.35 OD/mL in TYP or GG 5-5 medium and incubated at 20 °C. The inoculum was produced at 0.1 OD/mL and 20 °C in 50 mL TYP or GG 5-5 medium supplemented with Schatz salts in a 250 mL Erlenmeyer flask with shaking. The 20 °C growth curves were plotted for 72 h, where points were collected every 2 h. Regarding the 0 °C cultures, a single colony was inoculated in 3 mL TYP incubated at 15 °C overnight, and then diluted 0.4 OD/mL in TYP or GG 5-5 medium and incubated at 0 °C. The inoculum was produced at 0.2 OD/mL and 20 °C in 50 mL TYP or GG 5-5 medium supplemented with Schatz salts in a 250 mL Erlenmeyer flask with shaking. The 0 °C growth curves were plotted for about 200 h, where points were collected every 8 h. The experiment was set up with biological duplicates, in which the error was reported as the standard deviation determined by the Excel *dev.st.c* function.

A significance level of 0.05 was chosen prior to the experiments. Two-sided Student *t*-tests of two samples were conducted using GraphPad Prism 8 to test the null hypothesis that there would be no difference between means.

### 2.6. Segregation Stability and Incompatibility Assay

Plasmid segregation stability was assayed under non-selective conditions for over 150 generations. Each strain was selected from a single colony on TYP agar supplemented with 100 μg/mL ampicillin and grown in 3 mL TYP medium at 15 °C for 24 h with shaking. Then, dilution was performed in 3 mL GG 5-5 medium supplemented with Schatz salts and 100 μg/mL ampicillin, and the diluted culture was incubated for 24 h at 15 °C with shaking. The stability assay was performed by growing the bacteria at 15 °C in 5 mL GG 5-5 medium in non-selective conditions, inoculating at 0.05 OD_600_ until the exponential phase (1–2 OD). This step was performed daily until the desired number of generations.

The percentage of recombinant cells was assessed by seeding diluted samples on non-selective TYP agar. We diluted 0.1 OD/mL (1–2 × 10^7^ CFU/mL) culture at 1:10^4^, and 100 μL was seeded, obtaining about 100–200 colonies after incubation at 15 °C for three days. Then, 30–33 colonies were taken from each plate and replicated on both selective (100 μg/mL ampicillin) and non-selective plates, which were incubated at 15 °C for 24 h. Each culture was performed as a biological duplicate, and the average number of colonies resistant to 100 μg/mL ampicillin was calculated. The plasmid rate loss was calculated by the number of colonies grown on the TYP-agar plates with ampicillin divided by the number of colonies grown on the TYP-agar plates without ampicillin.

During the incompatibility assay, the *rep*-based replication plasmids were transferred to *Ph*TAC125 strains via intergeneric conjugation using *E. coli* strain S17-1(λpir) as a donor strain. The pMAV plasmid was a positive control for the intergeneric conjugation experiment, and *Psychrobacter* sp. TAD1 was a positive control for pUC-oriT-pTAUp plasmid intergeneric conjugation. The mating step was performed under non-selective conditions at the temperature of 15 °C. Selection of trans-conjugative clones was performed at the temperature of 4 °C using 100 μg/mL ampicillin. These selection conditions allowed us to specifically select for the psychrophilic recipient bacteria carrying the desired plasmid, as the mesophilic donor strain cannot grow or replicate at a temperature as low as 4 °C. Then, the clones were selected and subjected to replica plating using 100 μg/mL ampicillin.

### 2.7. Plasmid Copy Number Quantification

The relative plasmid copy number (PCN) was determined by the quantitative PCR (qPCR) technique. Amplification and data analysis were performed with a StepOne Real-time PCR System (Applied Biosystems, Foster City, CA, USA) along with the SYBR^®^ Green PCR Kit (Applied Biosystems, Foster City, CA, USA). The gene with locus tag *PSHA_RS10135* (alternatively known as *PSHAa2051*) was employed to detect chromosomal DNA in the samples, while the *ampR* gene was used for the detection of the plasmid. Each pair of primers (prom 7 fw, prom7 rv, BlaM fw, BlaM rv, Appendix A) (Eurofins Genomics, Ebersberg, Germany) was selected using the free Primer 3 web software [39].

For PCN estimation of unknown samples, the total DNA was extracted from 1 OD_600_ cell pellets of *Ph*TAC125 exponential-phase cultures grown in both TYP and GG 5-5 media. DNA extraction was performed using the Bacterial DNA kit, following the manufacturer’s instructions. DNA concentrations were measured with a NanoDrop TM 1000 Sp spectrophotometer (Thermo Fisher Scientific, Waltham, MA, USA) at the absorption of 260 nm, and purity was assessed by measuring the A260/280 and A260/230 ratios of the extracted DNA. For the PCN quantification, 10 ng DNA template was used. PCR reactions were prepared in 10 μL mixtures containing 1 × PowerUp SYBR Green Master Mix (Applied Biosystems, Foster City, CA, USA) with ROX as a passive reference dye and Uracil-DNA glycosidase (UDG) to eliminate contamination, 400 nM of each primer, and 1 μL of sample. The reaction master mixes were aliquoted in three wells of a reaction plate to perform technical replicates. Finally, the plate was sealed with an adhesive cover.

The thermal cycling protocol was as follows: UDG activation for 2 min at 50 °C; initial denaturation for 10 min at 95 °C; and 40 cycles of denaturation for 15 s at 95 °C alternated with annealing/extension steps for 1 min at 60 °C. Each reaction was performed in triplicate. Standard curves were developed using 10-fold serial dilutions of a random real sample (from 6 × 10^3^ to 6 pg of total DNA). In each dilution series, either the chromosomal gene or the plasmid gene was the target.

The amplification efficiency (E) of each gene was calculated from the slope of the relative standard curve (E = 10(−1/slope)). The relative PCN was estimated with the following equation: = (E_c_)^Ctc^/(E_p_)^Ctp^, considering the Ct values for the two amplicons (chromosome-c and plasmid-p) and the amplification efficiency of the plasmid (Ep) and chromosomal gene (Ec) [40].

### 2.8. H_2_O_2_ Disk-Inhibition and Motility Assay

The disk-diffusion assay was performed in the presence of 265 mM H_2_O_2_. A single colony of each strain was inoculated in 3 mL of TYP medium in non-selective conditions. Cells in the stationary phase were diluted to 0.2 OD/mL in 6.5 mL TYP soft agar (0.4% agar) and poured into a plate. Disks of Whatman filter paper (0.6 cm) were treated with 3 μL of 3.6% H_2_O_2_, placed in the center of each plate, and incubated at 15 °C. After 24 h of incubation, the area of inhibition was measured in cm. The experiment was performed as five independent replicates.

The motility assay was carried out for 72 h of incubation. Each strain was streaked on TYP agar and incubated at 15 °C for three days. A single colony was inoculated in 3 mL TYP medium and incubated at 15 °C for 24 h. Upon reaching the stationary phase, the cells were diluted at 2 OD/mL in TYP medium, and 25 µL were spotted on sterile Whatman filter paper placed at the center of a 0.3% soft agar TYP medium plate. After drying the spot in sterile conditions for 10 min, the plates were incubated at 15 °C. The length of the path (cm) was measured every 24 h. The experiment was performed as three technical replicates.

### 2.9. Biofilm Formation Assay

Each strain was grown in TYP medium and GG 5-5 medium at 15 °C and 0 °C for 144 h, and points were collected every 24 h. A single colony of each strain was inoculated in 2 mL of TYP at 15 °C for 24 h. Cells were diluted to 0.35 OD_600_/mL in 20 mL TYP media and GG 5-5 supplemented with Schatz salts. The pre-inoculum was split: 10 mL was incubated at 15 °C for 8 h with shaking in 100 mL Erlenmeyer flasks, and the other 10 mL was incubated at 0 °C for 72 h to perform culture temperature pre-adaptation.

For each temperature and media condition, the wells of a sterile, 24-well, flat-bottomed polystyrene plate were filled with 1 mL of a medium with a 0.2 OD_600_/mL dilution of the Antarctic bacterial culture in the exponential growth phase in static conditions. The plates were incubated either at 15 °C or 0 °C and the kinetics of biofilm formation were carried out for 24, 48, 72, 96, 120, and 144 h.

After rinsing with PBS, the adherent cells were stained with 0.1% (*w*/*v*) crystal violet, rinsed twice with double-distilled water, and thoroughly dried. Subsequently, the dye bound to the adherent cells was solubilized with 20% (*v*/*v*) acetone and 80% (*v*/*v*) ethanol. After 10 min of incubation at room temperature, the OD_590_ nm was measured to quantify the total biomass of biofilm formed in each well. The OD_590_ nm values reported were obtained by subtracting the OD_590_ values of the control obtained in the absence of bacteria. Each data point was composed of four independent samples.

### 2.10. Statistics and Reproducibility of Results

Data were statistically validated using the two-sided Student *t*-test comparing the mean values to the wild-type (wt) control. The significance of differences between the mean values was calculated using a two-sided Student *t*-test, and *p* < 0.05 was considered significant. For multiple comparisons, an ordinary one-way ANOVA test was performed using the Bonferroni test correction. A significance level of 0.05 was chosen prior to the experiments. Statistical analyses were conducted using GraphPad Prism 8.

## 3. Results

### 3.1. Genomic and Transcriptomic Profiling of the pMEGA Plasmid

In 2019, third-generation sequencing technologies led to the reannotation of the *Pseudoalteromonas haloplanktis* TAC125 genome, unveiling the presence of pMEGA. Previous NCBI nucleotide searches of this megaplasmid performed by Qi et al. with the NCBI [10] revealed only two significant hits to plasmids from *Pseudoalteromonas arctica* (36% of identity) and *Pseudoalteromonas nigrifaciens* (34% of identity), and scarce similarity with chromosomes I (5%) and II (2%) of *Ph*TAC125 [10].

Currently (October 2024), pMEGA nucleotide similarity searches against the NCBI nucleotide collection database (nr/nt) reveal two more hits, covering 33% and 26% of the pMEGA sequence, corresponding to endogenous plasmids of *Pseudoalteromonas* sp. KG3 and *Pseudoalteromonas* sp. PS1M3, respectively (Table 2).

The former hit matches the 184,073 bp long pPsKG3_1 plasmid from *Pseudoalteromonas* sp. KG3, isolated from cheese rind (NCBI:txid2951137) and showing a 95.04% identity with a region of pMEGA encoding for endonuclease subunit R (length: 5279 bp, coordinates: 221-3106, locus: NDQ71_24070) of the type I restriction–modification system, a gene encoding the toxin HipA (length: 1232 bp, coordinates: 1800-3032, locus: NDQ71_24235), and the *parA* gene (length: 1205 bp, coordinates: 30-1235, locus: NDQ71_24190).

The latter hit matches plasmid pPS1M3-2 (NZ_AP022626), which is a large plasmid of 27,013 bp harbored by *Pseudoalteromonas* sp. PS1M3, a marine bacterium isolated from sea sediment from the Boso Peninsula in Japan [41]. This plasmid shares 95.05% of identity with a region of pMEGA corresponding to *parA* (length: 1205 bp, coordinates: 30-1235, locus: PS1M3_39380) and *parB* genes (length: 1080 bp, coordinates: 1240-2322, locus: PS1M3_39390) (98% of identity taken individually), encoding the plasmid partitioning system, and the *hsdR*_2 gene (95% of identity), encoding a type I restriction endonuclease (length: 2888 bp, coordinates: 211-3099, locus: PS1M3_39320). Overall, the NCBI nucleotide BLAST analysis results showed hits against *Pseudoalteromonas* species, spanning from marine bacteria (i.e., *P. arctica* and *P. nigrifaciens*) to non-marine bacteria (i.e., *Pseudoalteromonas* sp. *KG3*). However, there were also significant matches with *Vibrio* species (CP100422.1, *Vibrio furnissii* chromosome I; CP089603.1, *Vibrio furnissii* strain VFN3 chromosome I), with an 8% query coverage and 87.44% identity, particularly in intergenic regions and pseudogenes (Table 2).

Protein similarity searches of pMEGA against the NCBI non-redundant protein sequences (nr) revealed that pMEGA-encoded proteins show similarity with two proteins, shared with *Pseudoalteromonas* sp. KG3 and *Pseudoalteromonas* sp. PS1M3. One hit corresponds to the ParA protein (WP_086998901, WP_024606458), sharing more than 99% identity, while the other is an AAA family ATPase protein (WP_301562706, WP_006793244), sharing more than 30% identity (Table 3).

Additionally, the 64,789 bp sequence of the pMEGA plasmid encompasses 52 annotated open reading frames (ORFs) classified into six different functional categories [10]. Following up on differential gene expression analyses that we conducted in a previous work for the same bacterium under two temperature conditions [37], we re-examined the transcriptional profile, focusing specifically on the genes harbored by pMEGA. Among the 50 encoded genes, we identified 27 differentially expressed genes, of which 2 genes and 25 genes were, respectively, down- and up-regulated at 15 °C (Table 4). The two genes (*PSHA_p00024*, *PSHA_p00035*) down-regulated at 15 °C were annotated as a DUF2059 domain-containing protein (PF09832) and HPP protein (PF04982), respectively, but their function is unknown or uncertain. The uncharacterized DUF2059 is a domain predominantly found in bacterial proteins. The HPP is an integral membrane protein, equipped with four transmembrane-spanning helices, which might function as a transmembrane transporter.

On the other hand, within the 25 genes up-regulated at 15 °C, 3 were annotated as hypothetical proteins (*PSHA_p00012*, *PSHA_p00037*, *PSHA_p00042*). Moreover, we identified six genes with higher up-regulation at 15 °C. Two of them, the *PSHA_p00001* and *PSHA_p00002* genes encoding the ParA and ParB proteins, respectively (PF18607, PF08775), regulate the plasmid segregation system of pMEGA. Notably, among the top five identified genes, *PSHA_p00022* and *PSHA_p00023* show the highest rate of up-regulation at 15 °C, which is 5-fold greater than that at 0 °C.

These genes represent the hybrid toxin–antitoxin systems encoding the YefM antitoxin (IPR036165) and the ParE toxin (IPR007712), respectively, involved in pMEGA maintenance and stability [10]. Taking this into consideration, the up-regulation of *par* genes and the toxin–antitoxin modules at 15 °C, the optimal growth temperature for the bacterium, might reflect the higher cellular replication rates and hence the increased demand for an efficient and robust partitioning system to ensure plasmid distribution to the daughter cells. Interestingly, the last gene (*PSHA_p00043*) encodes an AAA+ protein or ATPase, which engages in diverse activities such as acting as a molecular chaperone, ATPase subunit of protease, or helicase and translocating macromolecules or nucleic-acid-stimulated ATPases (PF13191). This protein might be involved in different stages of cellular growth, such as nutrient uptake, DNA replication, and repair.

Besides the identified differentially expressed genes (DEGs), it is worth noting that the *PSHA_p00052* gene is annotated as a replication initiation protein (RepB) (PF01051). The transcriptional level of this gene is not down- or up-regulated at the analyzed temperatures. This was expected considering that previous studies highlighted the product of the *repB* gene as a key player in the RCR mechanism, coding for the plasmid replication initiator, RepB, crucially ruling the duplication of pMEGA [10]. Considering that *repB* is an essential gene for plasmid maintenance, it stands out as a good candidate for plasmid curing.

### 3.2. Design of the Curing Strategy

Although pMEGA is endowed with multiple genes, none can be considered a useful selection marker to track down plasmid depletion. Thus, to achieve proper curing of pMEGA, a sequential genetic scheme was developed as follows: (a) gene inactivation by in-frame insertion through homologous recombination to insert an antibiotic selection marker, (b) gene silencing of plasmid replication elements by PTasRNA technology, and (c) selection of cured clones by following the loss of the selection marker.

The first step was designed to introduce the chloramphenicol selection marker within pMEGA, developing a novel mutant strain of *Ph*TAC125, named KrPL insPolV. Construction of the KrPL insPolV mutant was carried out by exploiting the pAT non-replicating plasmid (suicide vector, pAT-VS-HR*umuC*) to introduce the chloramphenicol resistance gene (*cmR*) into pMEGA. Also, the suicide vector comprises the 259 bp homology region of the *umuC* gene (HR*umuC*), which is a fragment of the *PSHA_00030* gene located on pMEGA (named A in Figure 2a) and the gene of *eGFP* under the *placZ* promoter (optimized for *Ph*TAC125). Homology recombination requires the mobilization of the suicide vector pAT-VS-HR*umuC* to the host cell *Ph*TAC125 KrPL, a strain devoid of the pMtBL plasmid [32], by intergeneric conjugation harnessing the *E. coli S17-1 λpir* strain [21,25,32] as the donor. This procedure allows for the recombination event in which pAT-VS-HR*umuC* inserts into the pMEGA plasmid in correspondence with the homology region (Figure 2b).

The suicide vector inserts into the *umuC* gene encoding the catalytic subunit of the error-prone DNA polymerase V [42], both disrupting the *umuC* gene and inserting the chloramphenicol selection marker.

The presence of the eGFP was meant to be used as a selection marker to easily visualize transconjugants through fluorescence directly on agar plates. However, given the low copy number of the pMEGA plasmid, fluorescence was not easily detectable, making the selection of mutants by this approach unfeasible. Thus, taking advantage of the *cmR* selection marker, the selection of transconjugating mutants was carried out by plating cells on kanamycin, which is an episomal resistance of KrPL, and chloramphenicol TYP solid medium plates incubated at 15 °C.

Four chloramphenicol-resistant colonies were selected, and detection of KrPL insPolV mutants was performed through PCR screening analyses using different pairs of primers (*Nde*I_eGFP fw, *Kpn*I_eGFP rv; 5′*umuC* fw, *oriC* rv—listed in Appendix A) and bacterial DNA as a template. The PCR analysis confirmed that all four positive clones exhibit the suicide vector inserted into the homology region, definitively achieving the KrPL insPolV (Appendix A). Subsequently, curing of pMEGA was achieved with the PTasRNA technology by designing paired termini antisense RNA against the *repB* gene.

The antisense sequence was designed to harbor the ribosome-binding site (RBS) and start codon of *repB* within a loop region accessible for the targeting of the transcript mRNA, as previously described by Lauro and co-workers (Appendix A) [25]. To ensure that the curing strategy did not induce any off-target effect, we performed a BLAST analysis of the *repB* antisense sequence against the *Ph*TAC125 whole genome, confirming the *repB* gene of the pMEGA plasmid as a unique target. The antisense mRNA sequence of *repB* (Appendix A) was cloned in the psychrophilic expression vector pB40-79-PTasRNA*lon*, an ampicillin-based derivative of pB40-79C-PTasRNA*lon* used in our previous work [25]. We exchanged asRNA*lon* with the asRNA*repB* sequence, achieving pB40-79-PTasRNA*repB* (Appendix A). Then, the pB40-79-PTasRNA*repB* expression vector was mobilized to the KrPL insPolV strain by intergeneric conjugation.

### 3.3. Selection of the pMEGA-Cured Clones

An interference step using PTasRNA*repB* was performed, cultivating KrPL insPolV cells in GG 5-5 media at 15 °C, and the synthesis of PTas*repB* was induced using IPTG. No considerable impact on the growth behavior of the KrPL insPolV mutant was detected upon antisense production (Appendix A). Thus, the selection of the putative cured clones was carried out by observing the loss of chloramphenicol resistance until 32 h after induction of antisense RNA synthesis on selective and non-selective TYP-agar plates. As anticipated, upon induction of the antisense RNA, pMEGA replication is interfered, resulting in the loss of the plasmid and the appearance of psychrophilic cells sensitive to chloramphenicol. As a control, KrPL insPolV cells were grown in the same experimental conditions but without IPTG induction. In both experiments, 200 independent clones were replicated on selective and non-selective solid media, and 4 chloramphenicol-sensitive clones were observed only in the pool of antisense RNA-treated cells. Putative-cured clones were subjected to PCR analysis to prove pMEGA plasmid depletion, using bacterial DNA as a template. This screening is designed to amplify three different fragments: (i) the *prom7* fragment (80 bp), which belongs to the *PSHAa_2051* gene of chromosome I (chrI) and represents a positive control of the *Ph*TAC125 strain, (ii) the *CDS49* fragment (100 bp), one of the pMEGA plasmid genes, and (iii) a *umuC* (259 bp)-specific fragment overlapping with the inserted suicide vector. The *CDS49* and *umuC* fragments were selected to reveal the absence of the pMEGA plasmid in the cured colonies.

In a first PCR colony screening amplifying *prom7* and *CDS49*, only two out of four clones proved to be putatively cured clones, numbered 147 and 153. These were subjected to DNA genome extraction, and a clean PCR analysis using the purified template was performed to confirm the curing of the pMEGA plasmid. Indeed, in both selected clones, the *prom7* fragment was successfully amplified, while the absence of amplification of *CDS49* and *umuC* confirmed the loss of the pMEGA plasmid (Appendix A). These clones are representative of a novel strain of *P. haloplanktis* TAC125 KrPL generally referred to as KrPL^2^. The applied curing procedure emphasizes the efficacy of the PTasRNA*repB* strategy in curing a plasmid whose replication mechanism is speculated to be based on rolling circle replication (RCR) mediated by the RepB initiator protein [10]. Additionally, it undoubtedly represents a validation of the PTasRNA conditional gene-silencing technology previously designed by Lauro et al. to interfere with *lon* gene expression, underscoring the versatility of the silencing system [25]. Lastly, the KrPL^2^ (clone 147) strain was subjected to culture propagation for about 150 generations and replicated on TYP agar in the presence and in the absence of ampicillin to select a clone spontaneously cured of the pB40-79-PTasRNA*repB* plasmid used for pMEGA depletion.

### 3.4. Growth Performances of the Novel Strain KrPL^2^

Considering the presence of housekeeping genes harbored on pMEGA (see Section 3.1), we sought to investigate the effect of the removal of such metabolic genes on the growth patterns of the KrPL^2^ strain. Thus, we focused on the evaluation of growth behavior of the novel strain, comparing its specific growth rate and cell biomass to the wt and the progenitor KrPL. The evaluation was performed at three different temperatures, 0 °C, 15 °C and 20 °C, either in the complex TYP or defined GG 5-5 media (Appendix A).

Overall, the specific growth rate (μmax, h^−1^) of the wt strain was significantly higher when compared to its derivatives KrPL and KrPL^2^ at 15 °C and 20 °C in TYP media (Table 5).

Instead, the specific growth rate in GG at 20 °C of KrPL^2^ was 0.363 ± 0.015 h^−1^, which was slightly higher than the 0.318 ± 0.011 h^−1^ of the KrPL strain (1.14-fold), possibly indicating an advantage of the strain without the pMEGA plasmid under these specific conditions (Table 5). Significant differences were not observed in the other tested conditions.

The doubling times reflected the above findings, where KrPL and KrPL^2^ showed moderately higher doubling times than the wt cells under most of the conditions. This indicated that the elimination of pMEGA in KrPL^2^ had not dramatically affected its specific growth rate.

Furthermore, in terms of biomass per liter (g_cdw_/L), no evident differences were found, except in GG 5-5 at 15 °C. Compared to the wt, KrPL and KrPL^2^ yielded a lower biomass concentration per liter, in which the latter reached the lowest biomass concentration (Table 5). The complex TYP medium supports higher growth rates and greater biomass accumulation than the defined GG 5-5 medium, underscoring the importance of nutrient availability. Overall, no significant differences were observed in growth behavior in the specific tested medium and temperature conditions. Loss of the pMEGA plasmid did not affect KrPL^2^ strains’ viability, instead preserving their similar growth behavior in respect to the progenitor. What stands out the most is that KrPL and KrPL^2^ differed from wt in each of the tested conditions, calling for further evaluation of these differences.

### 3.5. Evaluation of rep-Based Plasmid Replicability

pMEGA is a *rep*-based plasmid in which RCR is controlled by the RepB replication initiator protein [11]. Usually, plasmids that share one or more elements of their replication machinery may encounter a plasmid incompatibility phenomenon, as observed with other *rep*-based plasmids such as pKT240 [37] and pTAUp plasmids [43,44]. The former is a broad-host-range plasmid derived from the RSF1010 plasmid belonging to the IncQ incompatibility group [34], and the latter is an endogenous rolling-circle plasmid of *Psychrobacter* sp. TA144 [43]. Since it was previously observed in the wt and KrPL strains that these plasmids were not able to replicate (unpublished data), we assessed the applicability and replicability of *rep*-based plasmids in three different *Ph*TAC125 strains using the above-mentioned plasmids alongside a pMtBL-derived control plasmid.

The strains chosen for this analysis were the cured *Ph*TAC125 KrPL^2^, the progenitor *Ph*TAC125 KrPL, and *Ph*TAC125 wt. Each was conjugated with three different ampicillin-based resistance vectors: (i) the broad-host-range pKT240 plasmid (12,392 bp), (ii) pUC-oriT-pTAUp (4110 bp), and (iii) the psychrophilic pMAV (5009 bp). The latter was conceived as a positive control of the intergeneric conjugation experiment. Also, *Psychrobacter* sp. TAD1 was used as a positive control for intergeneric conjugation of pUC-oriT-pTAUp, which is derived from *Psychrobacter* sp. TA144 [43].

The mating step was performed at 15 °C overnight to accommodate the growth requirements of both the mesophilic donor and the psychrophilic recipient strains. The selection of putative trans-conjugative colonies was performed at the temperature of 4 °C with the ampicillin antibiotic. In this way, we created a selective condition that exclusively allows the growth of the psychrophilic recipient bacteria, which have successfully acquired the ampicillin-resistant plasmid. Indeed, this temperature-specific selection prevents the mesophilic donor strain from growing or replicating. However, trans-conjugative cells showed colony formation at very different times.

After antibiotic selection, the cells harboring either pKT240, pMAV, or pUC18-oriT-pTAUp were replica plated on TYP agar supplemented with ampicillin. Then, clones were inoculated in TYP medium supplemented with the antibiotic. Finally, the chosen clones were subjected to plasmid extraction, and DNA was analyzed through agarose gel electrophoresis, but none of the examined strains harbored the *rep*-dependent plasmids.

On the other hand, the positive control, pMAV, was successfully conjugated in the three strains, as well as the pUC-oriT-pTAUp plasmid in the positive control *Psychrobacter* sp. TAD1. The observation that pKT240 and pUC-oriT-pTAUp were not retained, despite clone growth on selective media, may be a result of spontaneous mutations that can occur under selective conditions.

Altogether, the results revealed that broad-host-range pKT240 and pUC18-oriT-pTAUp plasmids were not able to replicate in *Ph*TAC125 wt, KrPL, and KrPL^2^. This experiment thus excluded the involvement of the pMEGA replication machinery in the observed absence of replication of *repB*-based plasmids.

### 3.6. Evaluation of Plasmid Segregation Stability

Depletion of the pMEGA plasmid might vary the stability of a heterologous introduced plasmid (pMtBL-derived). Indeed, we found some genes on the pMEGA plasmid that might play a critical role in the segregation stability of the heterologous plasmid, such as a putative transposase, a restriction–modification system (RM), and toxin–antitoxin modules (TA) [44,45]. Thus, we evaluated the possible stability variability when such replicons were introduced into the KrPL^2^ strain, to assess the impact of pMEGA removal on plasmid stability. This study compared the loss rate of the high-copy-number psychrophilic plasmid pB40 [46] for over 150 generations in the presence and absence of pMEGA in non-selective conditions. We compared five strains: KrPL^2^ pB40-79-PTasRNA*repB* clone 147 and clone 153, the intermediate strain KrPL insPolV pB40-79-PTasRNA*repB*, and the progenitor KrPL pB40-79-PTasRNA*repB*.

Also, a control for the experiment was set up, exploiting the KrPL strain harboring the pMAI-79-*eGFP* low-copy-number plasmid, whose stability undergoes evident reduction over time in the percentage of recombinant cells (unpublished data). The results showed that the removal of pMEGA did not interfere with the stability of the pB40 high-copy-number plasmids in the KrPL^2^ cured strains (both clones 147 and 153) and the KrPL insPolV intermediate strain (Appendix A). The high-copy-number plasmids showed higher stability for over 150 generations (about 100%) in each examined strain compared to the low-copy-number plasmid, whose stability decreased from 90% after 79 generations to 65% after 150 generations (Appendix A). The elimination of the pMEGA plasmid did not affect the stability of introduced heterologous pB40 plasmids. Also, the deletion of the *umuC* gene (inactivation of DNA polymerase V) in the intermediate strain was not involved in plasmid segregational stability.

### 3.7. Assessment of Plasmid Copy Number

Similarly, we evaluated whether plasmid copy number (PCN) of introduced heterologous plasmids was affected. Quantitative PCR (qPCR) was performed to determine the PCN of pMAV, a psychrophilic low-copy-number plasmid, comparing KrPL^2^ to the wt and KrPL strains. We evaluated the PCN in both TYP and GG 5-5 media during the exponential growth phase (Table 6). The plasmid copy numbers estimated in KrPL^2^ were 4.04 ± 0.34 and 6.03 ± 0.53 in TYP and GG 5-5 conditions, respectively. However, the results did not show any differences compared to the wt and KrPL strain in the GG 5-5 medium, in which the estimated values were 4.68 ± 0.71 and 5.33 ± 0.69, respectively. Surprisingly, pMAV PCN in KrPL and the TYP medium resulted in 7.99 ± 0.67, which was 2-fold and 4-fold the PCNs, respectively, of KrPL^2^ and wt (2.61 ± 0.52).

The unique behavior of KrPL in TYP medium might be related to host cell factors, but further investigations are needed to understand this variability among the three strains. Overall, the PCN of heterologous plasmids was not affected by removal of the pMEGA plasmid in the KrPL^2^ strain.

### 3.8. Evaluation of Cellular Oxidative Stress and Motility

To further explore the phenotypes conferred by the elimination of pMEGA, we focused on the ability of KrPL^2^ to cope with cellular stress, evaluating its response to oxidative stress and swarming motility.

The sensitivity to oxidative agents was measured by a disk-diffusion assay using 265 mM hydrogen peroxide and comparing KrPL^2^, along with KrPL and wt cells, upon reaching the stationary phase. After 24 h of incubation at 15 °C, the growth of KrPL^2^ showed a significantly lower diameter of inhibition, suggesting its higher capacity to respond to oxidative stress induced by H_2_O_2_ (Table 7).

Swarming ability was evaluated through a TYP soft-agar motility assay, and parameters were expressed as the length of the path travelled on the TYP soft-agar plates per unit of time. The strain KrPL^2^ displayed the same ability to move in a semi-solid TYP medium at 15 °C compared to the wt and KrPL (Appendix A).

### 3.9. Biofilm Formation Kinetics

To investigate differences in the capacity for biofilm production of *Ph*TAC125 KrPL^2^ compared to the KrPL and wt strains, cells were grown in different media at 15 °C in static conditions. Bacteria were grown in TYP and GG 5-5 at 15 °C and 0 °C, and the biofilm was evaluated at different incubation times. The inoculum was produced at 0.2 OD_600_/mL in 20 mL TYP and GG 5-5 supplemented with Schatz salts in 24-well plates.

The kinetics of biofilm formation were carried out for 24, 48, 72, 96, 120, and 144 h. As shown in Figure 3, the kinetics of biofilm formation showed different patterns in KrPL^2^, wt, and KrPL at 15 °C in GG 5-5. Indeed, both KrPL and KrPL^2^ were not able to accumulate biofilm when grown in GG 5-5 medium at 15 °C after 24 h, though this was possible for wt. Furthermore, while KrPL biofilm accumulation gradually decreased over time, with a lower peak of accumulation at 144 h, KrPL^2^ showed a reduced ability to produce biofilm already after 48 h, which lasted until 144 h. KrPL and KrPL^2^ also showed differences in the production of biofilm in GG 5-5 medium at 0 °C after 24 and 48 h. However, after 48 h, differences between wt, KrPL, and KrPL^2^ were not evident.

Meanwhile, in the TYP complex medium, significant differences between the strains were not observed, except for the finding that wt accumulated less biofilm than the other strains at 15 °C after 144 h. Nevertheless, as previously described in *Ph*TAC125 wt, biofilms of KrPL and KrPL^2^ mainly accumulate at the air–liquid interface, forming pellicles at 15 °C in both TYP and GG 5-5 media [47]. Conversely, at 0 °C, *Ph*TAC125 biofilm mainly accumulates at the solid–liquid interface, and KrPL^2^ and KrPL begin to accumulate biofilm with a 24 h delay compared to wt, which is maintained until 96 h.

## 4. Discussion

Since *Pseudoalteromonas* is reported as a prevalent genus in marine ecosystems [48], studying its multipartite genome, particularly the presence of cryptic and mega-plasmids [29], may be of relevance for understanding its adaptation mechanisms to cold environments. In this regard, *P. haloplanktis* TAC125 represents a model microorganism for studying environmental adaptation mechanisms. Moreover, from a biotechnological perspective, the presence of the pMEGA plasmid may limit the potential of *Ph*TAC125 as a cell factory for difficult proteins.

In this study, genomic analyses revealed that genes encoding partitioning systems (*parA* and *parB*) and replication machinery (*repB*) are highly conserved across the genus (when present), with over a 90% sequence identity. The ParABS partitioning system is a tripartite mechanism comprising the ATPase protein ParA, the CTPase/DNA-binding protein ParB, and a centromere-like *parS* site. During cell division, ParB binds to the *parS* site, recruiting additional ParB molecules to form a protein–DNA complex, which activates the ATPase activity of ParA, driving the segregation of plasmids to opposite poles of the cell [49]. These systems, which rely on the active transfer of genetic material, are essential for the stability and retention of plasmids [50].

Transcriptomic analyses further support the stability of the pMEGA plasmid, revealing a high transcriptional activity of *parAB* partitioning genes, with notable up-regulation at 15 °C. This suggests that the efficient segregation and partitioning mechanism might be attributed to the ParAB system encoded by the *PSHA_00001* and *PSHA_00002* genes. Additionally, we speculate an addiction mode between the pMEGA plasmid and the toxin–antitoxin genes. TA modules, which are highly transcribed at 15 °C in *Ph*TAC125, are known to be responsible for plasmid persistence through a mechanism called “post-segregational killing of plasmid-free segregants”, which prevents plasmid loss [50]. In an addiction mode, *par* and TA module genes might be accountable for the high inherent stability of the pMEGA plasmid. However, further studies are necessary to address this putative mode and provide insights into megaplasmids’ role in psychrophilic bacteria.

To overcome the inherent stability of the endogenous pMEGA plasmid and successfully cure it from *Ph*TAC125, we applied a dual-step approach. This involved genetic manipulation of the pMEGA plasmid to insert a chloramphenicol resistance marker through homologous recombination, followed by interference with plasmid replication using paired termini antisense RNA (PTasRNA) technology. To streamline the curing procedure, we identified the *repB* gene (*PSHA_00052*) as a critical candidate, given its essential role in the plasmid replication machinery. After integrating the chloramphenicol resistance marker to facilitate the detection of plasmid loss, we proceeded to silence the *repB* gene using PTasRNA technology. This method has been shown to efficiently down-regulate chromosomal genes in *Ph*TAC125, achieving high levels of silencing [25]. Our strategy proved successful in curing the pMEGA plasmid, with 2 out of 200 screened colonies identified as mutants achieving the novel KrPL^2^ strain. This curing system validates the ability of the previously developed asRNA technology in *Ph*TAC125 to silence plasmid genes and to be applied for curing of megaplasmids. Yet, we observed a relatively low curing efficiency, which may be attributed to the high level of transcripts of the *repB* gene. As reported by Lauro et al., complete silencing of the *lon* gene, which is highly transcribed, was not observed, whereas complete silencing was achieved for the *PhhbO* gene, which is transcribed about 40 times less than *lon* [25]. The *repB* gene is transcribed only three times less than the *lon* gene, supporting the hypothesis that high levels of transcripts may hinder gene-silencing efficacy.

The KrPL^2^ strain exhibited a comparable specific growth rate to its progenitor, KrPL, across various temperatures and media, indicating that many genes encoded by the pMEGA plasmid are not essential under laboratory conditions. Its housekeeping genes, involved in replication (*repB*), partitioning (*parA* and *parB*), and plasmid maintenance (toxin–antitoxin systems), are responsible for its replication and persistence in cells without any effect on cell viability and without imposing a metabolic burden. Although seemingly non-essential in controlled environments, pMEGA might provide benefits in environmental settings where selective pressures such as resource unavailability, competition, or environmental stressors are predominant [51]. In the context of the biotechnological application of *Ph*TAC125 as a cell factory, potential differences may highlight the importance of the KrPL^2^ strain in recombinant protein production efficiency and productivity.

Interestingly, KrPL^2^ showed an enhanced tolerance to oxidative stress compared to both the wt and KrPL strains (Table 7), and biofilm formation was notably reduced in GG 5-5 medium at 15 °C, decreasing over time (Figure 3). Sequence similarity searches using pMEGA identified one gene that might be involved in the regulation of cellular stress, the TetR/AcrR transcriptional regulator (*PSHA_p00036*). This family of transcriptional factors acts as a repressor of transcription, regulating a wide range of cellular activities, including osmotic stress and homeostasis, and thus its members are defined as players in global cellular stress [52]. Interestingly, *acr* genes also play a role in biofilm formation, such as the deletion of *acrB* in *Salmonella* Typhimurium, which results in impaired biofilm formation [53]. Conversely, in *Acinetobacter nosocomialis*, *acrR* deletion causes an increase in biofilm/pellicle formation [54]. Given the potential role in the regulation of stress responses of the TetR/AcrR transcriptional regulator found on pMEGA, further studies are necessary to elucidate its specific contributions to the observed phenotypes in *Ph*TAC125. Additionally, the *Ph*TAC125 genome contains multiple putative *acr* genes, as reported by the MaGe Platform (MicroScope online resource) [55]. This suggests that *Ph*TAC125 might be equipped with a sophisticated and finely regulated system of Acr efflux pumps that likely contribute to various bacterial functions. Only further studies (such as gene deletion or the silencing of the *PSHA_p00036* gene) can elucidate the molecular mechanism underlying the role of the *PSHA_p00036* in this regulatory network.

Moreover, a significant portion of the genes on pMEGA (19 out of 50) are still lacking functional annotation. This raises the possibility that some of these uncharacterized genes could play critical roles in the regulation of the oxidative stress response and biofilm formation. The lack of comprehensive annotation makes it challenging to pinpoint the exact genetic determinants driving the observed phenotypic changes. Only targeted studies aimed at uncovering their specific functions will help us understand the molecular mechanisms contributing to these phenomena.

Further studies aiming to explore the mechanisms behind these pMEGA functions are of broad interest. Phenomena like a controlled stress response might be interconnected with environmental stressors that these organisms face in Antarctic habitats. Improved understanding will elucidate the adaptation mechanisms that influence microbial survival, community dynamics, and interactions. Furthermore, it is worthwhile mentioning that strains with enhanced tolerance to oxidative stress are highly appealing for industrial biotechnology. In large-scale industrial fermentation systems, oxidative stress can limit cell viability and productivity. Developing strains with improved tolerance has led to more robust and reliable production platforms, as demonstrated in *Clostridium tyrobutyricum* and *Saccaromyces cerevisiae* [56,57]. In the context of a reduced tendency for biofilm formation, they could simplify bioreactor maintenance, minimize contamination risks, and improve the overall process efficiency by reducing clogging or fouling of equipment [58].

Additionally, this study opens up the possibility of harnessing some of pMEGA’s valuable genetic elements, such as the origin of replication and/or promoters, and repurposing those for the construction of advanced genetic tools, to shed light on the exploitation of microorganisms from polar ecosystems for biotechnological purposes.

## 5. Concluding Remarks

In this study, we developed a novel strain of *P. haloplanktis* TAC125, named KrPL^2^, by successfully curing the megaplasmid pMEGA through a dual genetic approach combining homologous recombination and PTasRNA technology. The cured strain exhibited enhanced resistance to oxidative stress, which is a critical feature for biotechnological applications, especially in industrial fermentation systems where oxidative stress is a limiting factor. Additionally, the reduced capability of biofilm formation offers potential advantages in industrial settings by improving the process efficiency and reducing contamination risks. Finally, the streamlined genomic background of KrPL^2^ provides a robust platform for designing and testing new psychrophilic genetic tools leveraging the unique properties of psychrophilic bacteria’s genetic features.

In conclusion, our development of a robust curing strategy represents a significant achievement in building our understanding of the physiological role of megaplasmids in Antarctic bacteria and highlights potential biotechnological applications of *Ph*TAC125.

## Figures and Tables

**Figure 1 microorganisms-13-00324-f001:**
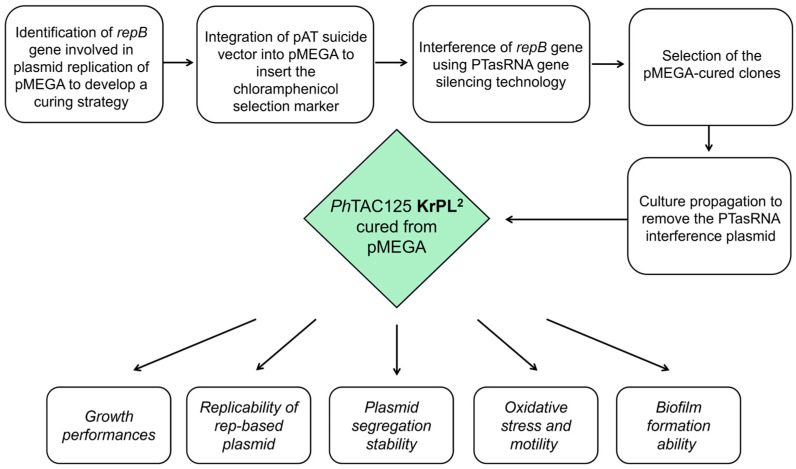
Workflow of the experimental design.

**Figure 2 microorganisms-13-00324-f002:**
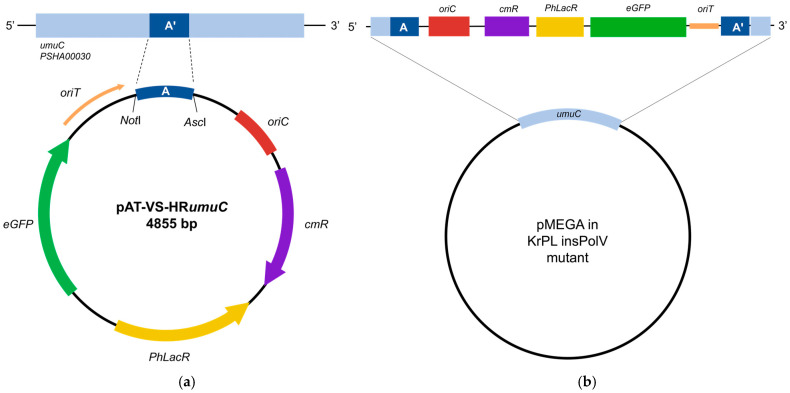
Insertion of the selection marker in pMEGA. (**a**) Schematic representation of the suicide vector. The *umuC* homology (259 bp) region A (dark blue) is inserted between the *Not*I and *Asc*I restriction sites; *oriC* is the origin of replication used to propagate the plasmid in *E. coli* strains (in red); *oriT* is responsible for the initiation of the conjugative transfer (in orange); *Ph*LacR encodes the regulator of the *placZ* promoter; *eGFP* represents the GFP reporter under the *placZ* promoter (in green); *cm*R represents the chloramphenicol resistance marker (in purple). (**b**) Representation of the inserted pMEGA plasmid in KrPL insPolV mutants. The suicide vector is inserted at the *umuC* gene (in light blue) of the pMEGA plasmid. The *umuC* is disrupted, gaining the *cmR* resistance marker and the eGFP reporter as key features.

**Figure 3 microorganisms-13-00324-f003:**
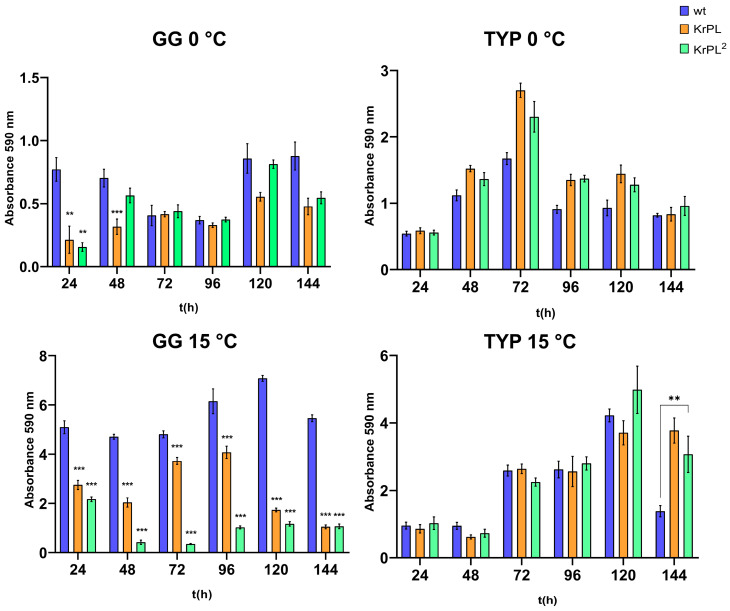
Analysis of the effect of the growth medium on the *Ph*TAC125 KrPL^2^ biofilm formation at different times. Comparison of *Ph*TAC125 wt, KrPL, and KrPL^2^ biofilms obtained at 15 °C and 0 °C in TYP medium and GG 5-5 medium. The biofilms were analyzed at 24, 48, 72, 96, 120, and 144 h with the crystal violet assay. Each data point was collected from four independent observations. Differences in the mean absorbance were compared to the wt strain samples according to Student’s *t*-test, with *p* < 0.01 considered significant (** *p* < 0.01; *** *p* < 0.001).

**Table 1 microorganisms-13-00324-t001:** List of plasmids used in this study.

Plasmid	Relevant Characteristics	Source
pMAV	Low copy number, pMtBL-derived, Amp^R^	[9]
pKT240	Broad host range RK2, Amp^R^	[34]
pUC-oriT-pTAUp	pUC18-derived, containing pTAUp origin of replication, Amp^R^	[32]
pAT-*eGFP*	Suicide plasmid, Amp^R^	[23]
pAT-VS-HR*umuC*	Suicide plasmid, Cam^R^	This work
pB40-79-PTasRNA*lon*	pMtBL-derived, containing *lon* asRNA, Amp^R^	This work
pB40-79C-PTasRNA*lon*	pMtBL-derived, containing *lon* asRNA, Cam^R^	[25]
pB40-79-PTasRNA*repB*	pMtBL-derived, containing *repB* asRNA, Amp^R^	This work

**Table 2 microorganisms-13-00324-t002:** BlastN similarity searches against the NCBI nr/nt nucleotide database.

Description	Query Coverage	E-Value	Identity (%)	Accession
*Pseudoalteromonas* sp. KG3 plasmid pPsKG3-1	33%	0.0	95.04	CP098525.1
*Pseudoalteromonas* sp. PS1M3 plasmid pPS1M3-2	26%	0.0	95.05	AP022626.1
*Vibrio furnissii* strain 104486766 chromosome I	8%	0.0	87.44	CP100422.1
*Vibrio furnissii* strain VFN3 chromosome I	8%	0.0	87.44	CP089603.1
*Pectobacteriaceae bacterium* CE90 chromosome	8%	0.0	87.59	CP129114.1

**Table 3 microorganisms-13-00324-t003:** Significant results of BlastP similarity searches of the proteins residing on pMEGA against the nr database.

Description	Protein Annotation	Query Coverage	E-Value	Identity (%)	Accession
*Pseudoalteromonas* sp. KG3 plasmid pPsKG3-1	ParA family protein	100%	0.0	99.75	WP_086998901
	Type I restriction endonuclease subunit R	100%	0.0	96.75	WP_301563121
	AAA family ATPase protein	89%	2.00 × 10^−53^	32.61	WP_301562706
*Pseudoalteromonas* sp. PS1M3	ParA family protein	100%	0.0	99.75	WP_024606458
plasmid pPS1M3-2	AAA family ATPase protein	91%	5.00 × 10^−55^	32.36	WP_006793244

**Table 4 microorganisms-13-00324-t004:** pMEGA differentially expressed genes (DEGs) at 15 °C and 0 °C.

Locus_Tag	Annotation	Log2FC	Padj Value	Protein ID
*PSHA_p00001*	ParA ATPase protein	0.85	1.41 × 10^−3^	WP_089369339
*PSHA_p00002*	ParB DNA-binding protein	0.91	1.78 × 10^−3^	WP_181718171
*PSHA_p00003*	TnpB transposase domain-containing protein	1.64	1.39 × 10^−8^	WP_011790088
*PSHA_p00010*	TonB-dependent siderophore receptor	1.06	1.38 × 10^−4^	WP_258557454
*PSHA_p00011*	Aminotransferase class V-fold PLP-dependent enzyme	0.68	4.73 × 10^−2^	WP_197108802
*PSHA_p00012*	Hypothetical protein	1.06	1.70 × 10^−2^	WP_181718177
*PSHA_p00014*	DNA replication terminus site-binding protein	0.92	1.67 × 10^−3^	WP_181718178
*PSHA_p00022*	Phd/YefM antitoxin type II toxin–antitoxin system	1.71	1.27 × 10^−5^	WP_181718184
*PSHA_p00023*	RelE/ParE toxin type II toxin–antitoxin system	2.00	2.89 × 10^−7^	WP_181718185
*PSHA_p00024*	DUF2059 domain-containing protein	−0.92	2.28 × 10^−3^	WP_181718186
*PSHA_p00026*	Tyrosine-type recombinase/integrase	0.55	4.75 × 10^−2^	WP_089369352
*PSHA_p00029*	S8 family serine protease	0.89	1.08 × 10^−3^	WP_181718190
*PSHA_p00031*	LexA family transcriptional regulator	1.14	2.47 × 10^−2^	WP_007583926
*PSHA_p00035*	HPP family protein	−0.77	1.51 × 10^−2^	WP_181718157
*PSHA_p00037*	Hypothetical protein	1.21	2.84 × 10^−4^	WP_181718158
*PSHA_p00040*	IS66 family insertion sequence element accessory protein TnpB	1.69	3.95 × 10^−6^	WP_181718161
*PSHA_p00041*	IS66 family insertion sequence hypothetical protein	1.85	1.45 × 10^−4^	WP_309477185
*PSHA_p00042*	Hypothetical protein	1.19	4.86 × 10^−3^	WP_181718162
*PSHA_p00043*	AAA family ATPase-binding protein	1.90	6.67 × 10^−10^	WP_181718163
*PSHA_p00045*	NAD-dependent oxidoreductase	0.73	2.26 × 10^−2^	WP_181718165
*PSHA_p00046*	Type I restriction–modification system, subunit R (restriction)	1.02	3.01 × 10^−5^	WP_181718166
*PSHA_p00047*	RNA-binding domain-containing protein	1.27	1.83 × 10^−7^	WP_181718167
*PSHA_p00048*	Type I restriction–modification system subunit S (specificity)	0.77	1.41 × 10^−2^	WP_181718168
*PSHA_p00049*	Type I restriction–modification system subunit M (modification)	0.91	5.44 × 10^−4^	WP_181718169

**Table 5 microorganisms-13-00324-t005:** Growth kinetic parameters.

µmax (h^−1^)	TYP			GG 5-5		
	0 °C	15 °C	20 °C	0 °C	15 °C	20 °C
wt	0.0775 ± 0.0088	0.457 ± 0.023	0.697 ± 0.051	0.0345 ± 0.0035	0.265 ± 0.025	0.529 ± 0.011
KRPL	0.0567 ± 0.0048	0.333 ± 0.007	0.482 ± 0.032	0.0310 ± 0.0019	0.222 ± 0.006	0.318 ± 0.011
KRPL^2^	0.0609 ± 0.0105	0.327 ± 0.006	0.483 ± 0.019	0.0320 ± 0.010	0.230 ± 0.013	0.363 ± 0.015
g, doubling time (h)	TYP			GG 5-5		
	0 °C	15 °C	20 °C	0 °C	15 °C	20 °C
wt	9.02 ± 1.01	1.52 ± 0.08	1.00 ± 0.07	20.20 ± 1.99	2.64 ± 0.31	1.31 ± 0.03
KRPL	12.26 ± 0.99	2.08 ± 0.04	1.44 ± 0.09	22.42 ± 1.35	3.13 ± 0.09	1.85 ± 0.26
KRPL^2^	11.61 ± 1.93	2.12 ± 0.04	1.44 ± 0.06	23.74 ± 7.90	3.02 ± 0.16	1.91 ± 0.08
Maximum biomass concentration (gdcw/L)	TYP			GG 5-5		
	0 °C	15 °C	20 °C	0 °C	15 °C	20 °C
wt	14.22 ± 0.30	15.45 ± 1.91	12.01 ± 1.21	5.31 ± 0.27	4.73 ± 0.1	3.95 ± 0.02
KRPL	12.98 ± 0.19	10.81 ± 0.46	11.15 ± 0.28	4.86 ± 0.14	4.09 ± 0.23	3.93 ± 0.33
KRPL^2^	13.68 ± 0.36	11.24 ± 1.04	11.33 ± 0.59	4.59 ± 0.29	3.46 ± 0.12	3.95 ± 0.28

**Table 6 microorganisms-13-00324-t006:** Plasmid copy number assay of the pMAV plasmid.

Strains	TYP	GG 5-5
wt	2.61 ± 0.52	4.68 ± 0.71
KrPL	7.99 ± 0.67	5.33 ± 0.69
KrPL^2^	4.04 ± 0.34	6.03 ± 0.53

**Table 7 microorganisms-13-00324-t007:** Oxidative stress assay.

Sample	Inhibition Diameter (cm)	*p*-Value
wt	0.5 ± 0.17	
KrPL	0.4 ± 0.07	not significant
^1^ KrPL^2^	0.22 ± 0.04	0.008

^1^ Differences in the mean values were compared to the wt control and considered significant when *p* < 0.05 according to Student’s *t*-test.

## Data Availability

The data presented in this work are available upon request from the corresponding author.

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
