# Peer review of "Engineering the Marine Pseudoalteromonas haloplanktis TAC125 via the pMEGA Plasmid Targeted Curing Using PTasRNA Technology"

_microorganisms, 2025, doi:10.3390/microorganisms13020324_

Round 1
Reviewer 1 Report
Comments and Suggestions for Authors
Specific comments
1. In reference to the assay conducted to determine the expression of the PTasRNArepB in the clones which do not replicate on TYP agar supplemented with chloramphenicol, I think that the PCR colony screening assay only show the presence of the plasmid that carries the PTasRNA interference, but not its expression. So, I think that it is necessary to probe the expression of this PTasRNA Interference by TR-qPCR or northern blot assays.
2. In material and methods, it is necessary to describe the preadaptation conditions used for the preinoculum.
3. In the incompatibility assay is not clear why was used 4°C to the selection of trans-conjugative clones, explain.
4. In the biofilm assays, the authors mentioned use an OD of 590 to conduce the assay, clarify which OD was used to determine the strains growth.
5. In the relative plasmid copy number determination, is important to clarify the concentration of chromosomic or plasmidic DNA used in the qPCR assays.
6. Clarify the length of the region, as well as the coordinates of pMEGA ORFs that showed the identity of the plasmids identified by Blast that made a match, since the description of the analysis is very general and the size of the regions is not clear. The authors only describe some of the genes located in these regions.
7. I think, that could be important that the authors shown the analysis of the regulatory regions to all these genes, since these genes showed the same change (up or down regulation) in their expression by the temperature employed. This analysis could complement the information of the expression and suggest regulation by a common cis element regulatory.
8. As was mentioned previously, the authors do not show evidence that the PTasRNArepB interference is expressed, in figure S4 shown that the induction with IPTG do no modify the strain growth, however, there is no evidence that, they have to demonstrate that the PTasRNArepB interference is produce.
9. It’s not clear, why the conjugation assay to the evaluation of plasmid segregation incompatibility was made at 4oC. Do the authors used another temperature (15oC) to conduce the experiment?. It could be necessary to use another condition to support the suggestion that pMEGA does’ not participate in the incompatibility to others plasmids. Furthermore, how do the authors explain the presence of selected colonies in ampicillin (supposed transconjugants), please clarify. It would be recommended that the authors perform a PCR analysis to identify some of the genes present in each plasmid used, to provide more evidence.
10. It is not clear why the authors would expect that the genes present in pMEGA could participate in the stability and copy number of heterologous plasmids, please clarify. To demonstrate the participation of genes located in pMEGA in the stability of other plasmids, it could be advisable to generate a mutation in these genes (such as the toxin-antitoxin system) and analyze the stability of the plasmids.
11. Correct the figure legend of figure 2, since the graphics not shown an statistic difference whit one asterix (*), and rewrite “The biofilms were analyzed at 24 h, 48 h, 72 h, 96, 120, and 706 144 h with the crystal violet assay.”
Minor comments
Line 133, is important to write the complete name the first time when is described a strain used.
Line 328 and 351, use the same style in the manuscript to describe “hours”.
Line 329, change “disk of inhibition” by “area of inhibition”.
Line 334, the cultures were diluted at 2 OD/mL in TYP medium?.
Lines 254 and 346, use the same redaction style to “pre-adaptation”.
Line 688, does not exist a “Table S5” in supplementary material, correct it by “Table S3”.
Author Response
Dear Reviewer,
Thank you very much for your detailed and insightful review of our manuscript. We sincerely appreciate your thorough evaluation, which has provided us with valuable suggestions to improve the quality and clarity of our work. We are grateful for the opportunity to address your concerns, and we highlighted the corresponding amendments in purple over the manuscript.
Comment 1: In reference to the assay conducted to determine the expression of the PTasRNArepB in the clones which do not replicate on TYP agar supplemented with chloramphenicol, I think that the PCR colony screening assay only show the presence of the plasmid that carries the PTasRNA interference, but not its expression. So, I think that it is necessary to probe the expression of this PTasRNA Interference by TR-qPCR or northern blot assays.
Response 1: (lines 528-539) Thank you for your thoughtful review regarding our experimental methodology. Your feedback has prompted us to carefully re-examine the original paragraph of the interference and clarify our methodology. We acknowledge that we might have been unclear about the experimental design.
Specifically, we first induced the antisense RNA targeting the repB gene to interfere with pMEGA replication. As a consequence, pMEGA is depleted and the chloramphenicol resistance marker is lost. Thus, we systematically selected for chloramphenicol-sensitive colonies, which serves as a direct functional indicator of plasmid loss. To confirm this, the PCR screening was strategically designed to confirm this plasmid elimination by both verifying the presence of a chromosomal gene prom7 (positive control) and confirming the absence of pMEGA-related genes CDS49 and umuC. In the light of the lack of clarity of the paragraph, we have thoroughly revised it to explicitly delineate these experimental objectives and validation strategies at lines 528-539. We appreciate the opportunity to clarify our experimental approach.
Comment 2: In material and methods, it is necessary to describe the preadaptation conditions used for the preinoculum.
Response 2: (lines 263-273) Thank you, the paragraph was rephrased as follows: “Regarding the 20 °C, a single colony is inoculated in 3 mL TYP incubated at 15 °C overnight. Following the culture is diluted 0.35 OD/mL in TYP or GG 5-5 media and incubated at 20 °C. The inoculum was performed at 0.1 OD/mL at 20 °C in 50 mL TYP or GG 5-5 medium supplemented with Schatz salts in 250-mL Erlenmeyer flask with shaking. The 20 °C growth curves were carried out for 72 h and points were collected every 2 h. Regarding the 0 °C cultures, a single colony is inoculated in 3 mL TYP incubated at 15 °C overnight, and then diluted 0.4 OD/mL in TYP or GG 5-5 media and incubated at 0 °C. The inoculum was performed at 0.2 OD/mL at 20 °C in 50 mL TYP or GG 5-5 medium supplemented with Schatz salts in 250-mL Erlenmeyer flask with shaking. The 0 °C growth curves were carried out for about 200 h and points were collected every 8 h.” See L263-L273.
Comment 3: In the incompatibility assay is not clear why was used 4°C to the selection of trans-conjugative clones, explain.
Response 3: (lines 301-305) Thank you for your comment regarding the use of 4°C in the incompatibility assay. The conjugation selection was performed at 4°C to facilitate the selective growth of trans-conjugative clones. This selection relies on the ability of the recipient psychrophilic bacteria to replicate at a temperature as low as 4 °C, whereas the mesophilic donor strain cannot. The selection at 4 °C and using the ampicillin antibiotic enable us to only select the psychrophile recipient carrying the desired plasmid. We included this explanation in the revised manuscript to clarify the rationale behind this approach. See L301-L305.
Comment 4: In the biofilm assays, the authors mentioned use an OD of 590 to conduce the assay, clarify which OD was used to determine the strains growth.
Response 4: (line 358) Thank you. The OD600 used to determine strain growth was specified in the new version of the manuscript. See line 358.
Comment 5: In the relative plasmid copy number determination, is important to clarify the concentration of chromosomic or plasmidic DNA used in the qPCR assays.
Response 5: (line 322) Thank you, done. You can find the adjustment at line 322.
Comment 6: Clarify the length of the region, as well as the coordinates of pMEGA ORFs that showed the identity of the plasmids identified by Blast that made a match, since the description of the analysis is very general and the size of the regions is not clear. The authors only describe some of the genes located in these regions.
Response 6: (lines 396-407) We thank you for this insightful comment. We have rephrased the relevant section to include the length of the aligned regions, as well as the specific coordinates of the identified ORFs. The updated description can be found between L396-L407.
Comment 7. I think, that could be important that the authors shown the analysis of the regulatory regions to all these genes, since these genes showed the same change (up or down regulation) in their expression by the temperature employed. This analysis could complement the information of the expression and suggest regulation by a common cis element regulatory.
Response 7: Thank you very much for your insightful suggestion, which is much appreciated. We agree that genes exhibiting the same change in expression under the employed temperature conditions could potentially be regulated by a common cis-regulatory element. This is a fascinating hypothesis and one that would certainly merit further investigation. However, we did not investigate this aspect since is not the main focus of this work. We acknowledge that only targeted studies specifically designed to analyze cis-regulatory elements and their influence on pMEGA gene expression would be able to highlight such molecular mechanism. This important point that you have raised inspired as to pursue other future research directions.
Comment 8: As was mentioned previously, the authors do not show evidence that the PTasRNArepB interference is expressed, in figure S4 shown that the induction with IPTG do no modify the strain growth, however, there is no evidence that, they have to demonstrate that the PTasRNArepB interference is produce.
Response 8: (lines 528-539) Thank you for your suggestions. As you mentioned previously, the proposed RT-qPCR or Northern blot experiments would serve to further confirm the production of PTasRNArepB. An indirect evidence of PTasRNArepB production and function is the pMEGA-curing outcome, in comparison with the control, i.e. the same experiment carried out without inducing the PTasRNA production. Therefore, we clarified better the experimental design to highlight the relevance of the control experiment. Indeed, the control experiments produced no chloramphenicol-sensitive cells and this strongly suggests specific and effective antisense RNA production against pMEGA in the curing experiment. We are concerned that conducting the RT-qPCR or Northern blot experiments asked by the reviewer requires some time and ends-up in a further publication delay of this manuscript. However, if the reviewer considers them essential, we would be available to carry them out. We hope we have addressed your concerns, trying to remain committed to transparency of our work.
Comment 9. It’s not clear, why the conjugation assay to the evaluation of plasmid segregation incompatibility was made at 4oC. Do the authors used another temperature (15oC) to conduce the experiment?. It could be necessary to use another condition to support the suggestion that pMEGA does’ not participate in the incompatibility to others plasmids. Furthermore, how do the authors explain the presence of selected colonies in ampicillin (supposed transconjugants), please clarify. It would be recommended that the authors perform a PCR analysis to identify some of the genes present in each plasmid used, to provide more evidence.
Response 9: (lines 597-636) Thank you for your valuable observation. Upon reviewing the text, I recognize that our phrasing might have unintentionally led to confusion regarding the purpose of the selection conditions. I have rephrased the paragraph to reflect the actual intent of conducting the experiment at the temperature of 4 °C, instead of 15 °C. I hope that this rephrasing makes the intention clearer and better conveys our goal. You can find the adjustment at L597-L636. Regarding the PCR methodology, while PCR analysis of specific genes within the plasmids could provide further confirmation, we believe that the detection of the plasmids themselves through gel electrophoresis provides solid evidence for the presence of the constructs.
Comment 10: It is not clear why the authors would expect that the genes present in pMEGA could participate in the stability and copy number of heterologous plasmids, please clarify. To demonstrate the participation of genes located in pMEGA in the stability of other plasmids, it could be advisable to generate a mutation in these genes (such as the toxin-antitoxin system) and analyze the stability of the plasmids.
Response 10: (lines 640-646) Thank you very much for your thoughtful comment. We acknowledge that we were not entirely clear in the original text, and we have rephrased the paragraph to improve clarity on why we expect pMEGA involvement in the stability at lines 640-646. We also agree with your suggestion that targeted studies would be necessary to determine if specific features, such as transposase gene or toxin-antitoxin system located on pMEGA, are responsible for plasmid stability/instability. However, it is important to clarify that the focus of this study was not to identify specific genes contributing to stability but rather to assess whether the heterologous plasmids were able to maintain the same level of stability upon pMEGA removal. Indeed, the observation that plasmid stability remains unchanged is particularly significant, as it highlights the potential of this strain to be implemented at an industrial level. In any case, we greatly appreciate your suggestions, as it opens interesting avenues for further exploration of the molecular mechanisms underlying plasmid stability and copy number.
Comment 11: Correct the figure legend of figure 2, since the graphics not shown an statistic difference whit one asterix (*), and rewrite “The biofilms were analyzed at 24 h, 48 h, 72 h, 96, 120, and 706 144 h with the crystal violet assay.”
Response 11: Thank you for your observation. We have adjusted the legend of the figure.
Minor comments
Comment 12: Line 133, is important to write the complete name the first time when is described a strain used.
Response 12: Thank you, done.
Comment 13: Line 328 and 351, use the same style in the manuscript to describe “hours”.
Response 13: Thank you, done.
Comment 14: Line 329, change “disk of inhibition” by “area of inhibition”.
Response 14: Done. Thank you.
Comment 15: Line 334, the cultures were diluted at 2 OD/mL in TYP medium?.
Response 15: Yes, thank you. I have adjusted.
Comment 16: Lines 254 and 346, use the same redaction style to “pre-adaptation”.
Response 16: Done. Thank you. The preadaptation style was preferred.
Comment 17: Line 688, does not exist a “Table S5” in supplementary material, correct it by “Table S3”.
Response 17: Thank you for pointing out this typo. It was corrected
Reviewer 2 Report
Comments and Suggestions for Authors
In the manuscript, the authors have conducted a comprehensive study on the endogenous giant plasmid pMEGA from the marine bacterium Pseudoalteromonas haloplanktis TAC125 (PhTAC125), shedding light on its functional role within the strain. The study demonstrates the successful elimination of the pMEGA plasmid, leading to the development of a new strain, KRPL2. Notably, KRPL2 exhibits no significant deviation in growth behavior from its progenitor strain but shows intriguing alterations in antioxidant stress response and biofilm formation capabilities. The innovative application of PTasRNA gene silencing techniques to eliminate a rolling circle replication plasmid is a highlight of this research. The findings significantly contribute to the understanding of plasmid functions in PhTAC125, particularly in the context of oxidative stress resistance and biofilm formation. Generally, the whole manuscript is well presented. All the claims and objectives of the study align well with the scope of Microorganisms. This manuscript would be suitable for publication following minor revisions.
Comments:
1. Whether the authors could provide a schematic overview or flowchart that encapsulates the study's design and workflow?
2. In lines 228-229, the authors employed IPTG at a concentration of 10 mM when the cell density reached 1.5 OD/mL. The authors should elucidate the rationale behind these specific parameters, including any prior experiments or literature that informed this decision.
3. While the manuscript documents changes in antioxidant stress and biofilm formation in the KrPL2 strain, there is a need for a more in-depth exploration of the underlying molecular mechanisms. The authors might provide potential explanations for these phenotypic changes in the the Discussion section.
Author Response
Dear Reviewer,
Thank you for your thoughtful and constructive feedback on our manuscript. We sincerely appreciate the time and effort you have dedicated to reviewing our work. We are pleased that you found our study to be comprehensive and well-presented, and we value your recognition of the significance of our findings. Regarding the minor revisions you mentioned, we are more than willing to address these points to further improve the clarity and accuracy of the manuscript. All the adjustments are highlighted in green in the text of the manuscript.
Comments 1: Whether the authors could provide a schematic overview or flowchart that encapsulates the study's design and workflow?
Response 1: (lines 128-129) Thank you very much for your precious suggestion. We have added a flowchart of the study design at the end of the Introduction as Figure 1 between L128-129.
Comments 2: In lines 228-229, the authors employed IPTG at a concentration of 10 mM when the cell density reached 1.5 OD/mL. The authors should elucidate the rationale behind these specific parameters, including any prior experiments or literature that informed this decision.
Response 2: (lines 233-235) Thank you for your suggestion. The rationale behind the usage of these parameters is based on the literature about the PTasRNA technology specifically developed for PhTAC125. The reference to the authors was added.
Comments 3: While the manuscript documents changes in antioxidant stress and biofilm formation in the KrPL2 strain, there is a need for a more in-depth exploration of the underlying molecular mechanisms. The authors might provide potential explanations for these phenotypic changes in the the Discussion section.
Response 3: (lines 802-818) Thank you very much for your thoughtful comment. I understand that this paragraph may have come across as unclear. Your feedback helped me recognize where the original phrasing might have been missing the point. Thus, to address the matter, I have rephrased the paragraph to streamline the content and make it more clear while maintaining its intended meaning. The adjusted version can now be found between lines L802 and L818.
Reviewer 3 Report
Comments and Suggestions for Authors
The highly detailed manuscript by Severino and colleagues reports on the curing of the large “pMEGA” plasmid from the psychotropic Pseudoalteromonas haloplanktis. This manipulation allowed the characterization of the role of the plasmid in growth adaptation, oxidative stress and motility. Overall the research is of interest and should allow further streamlining on this potential “cell-factory” as well as the exploitation of various plasmidogenic sequences as additional genetic tools. Most of the comments that I have listed below relate to language suggestions or clarification of meaning.
Specific comments
1. L22 and L26 I recommend using “of” rather than “from” e.g. L26 cured of the pMEGA plasmid
2. L29 of the pMEGA
3. L31 to exploit valuable pMEGA genetic elements and further
4. L43 and L58 appear top conflict. The temp range is given as 20-30 in L43 but then 15 degrees L58. Please clarify in the text.
5. L45, L46, L94 Gamma (capitalized and italicized); please check throughout ms
6. L61 a) a source
7. L62 [13,14], b)
8. L62 and c) an exceptional
9. L73 chromosomal gene expression (it may seem counterintuitive since more than one gene would be expressed, but this is the standard style of referring to such “plurals”, for greater fluency).
10. L79 bacteria, including Pseudoalteromonas
11. L86 vs L127 variable capitalization (and hyphenation) of rolling circle replication
12. L87 pMEGA encodes two type II
13. L88 YefM
14. L134 vs L136 different styles are used for “R” referring to resistance
15. L139 experiment. The Ph…
16. L140 was this mutant made in current study or was it already available. Please clarify.
17. L150 and throughout. I would write this as “Schatz salts” as written L701.
18. L156 through the heat-shock
19. L171-2 Please define NCBI and NIH on first use; also CDS on L180, L199 MCS
20. L184 Please provide a reference for gateway cloning procedures used
21. L186 Which PCR enzyme? Phusion (as listed L206)?
22. L187 a pair of forward and reverse oligonucleotides listed (also L207)
23. L206 and throughout; please provide company location on first mention. Please also include source of primers.
24. L211 used in this work
25. L217 and throughout 100-mL Erlenmeyer
26. L220 agar plates
27. L222 25-mL inoculum
28. L225 please define OD and provide wavelength on first mention
29. L229 Please make sure IPTG is defined on first use
30. L229 and throughout. The authors use “hours” (in full) and then “h” either with a space or without. Please use one style throughout. I recommend using 8 h (for example)
31. L231 and seeded on
32. L233 replicated on a 70-mL
33. L237 clones were subjected
34. L239 for the PCR using
35. L242 Growth Culture Conditions.. (please see note 9 above)
36. L244 on a TYP
37. L246 in 25-mL
38. L248 in a 100-mL (also L250, L344); L252 in a 250-mL
39. L256 culture set up
40. L257 before inoculation
41. L260 duplicates
42. L262 vs section 2.10 Three different styles are used for writing the t-test. Please use one.
43. L287 plasmid intergeneric (please see note 9)
44. L294, L302, L304, L307, L311 Company locations can be removed as given earlier
45. L296 ampR (no capitalization)
46. L297 Each pair of primers
47. L298 Please give a reference or source of Primer 3
48. L298 Table S1)
49. L306 10-uL mixtures
50. L329 the zone of inhibition was measured in cm. The experiment was
51. L336 10 min
52. L362 has “wt” been defined?
53. L383 184,073-bp long
54. L392 to the parA
55. Table 3. It is unclear why the hsd proteins were not identified by BlastP if they were found by Blastn
56. L413 encoded genes
57. L475 with multiple genes
58. L484 259-bp
59. L486, L498 vs L827 Three different styles used for eGFP
60. L469 RCR
61. L486, L498 different styles are used for placZ (I would use the style L486)—may have been an earlier example of PlacZ (please check)
62. L490 I would combine these 2 paragraphs
63. L503 inserts into the umuC (please check italicization of “C” in L504 also)
64. L509 selection of mutants by this approach, not feasible.
65. L509 vs L501 CamR or cmR
66. L514 different pairs
67. L517 vector inserted
68. L519, L521, L539 three different styles are used for writing antisense RNA. Please unify throughout ms.
69. L521 Please define RBS on first use
70. L525 against the
71. L547 in the pMEGA
72. L554 what is “clean PCR”? please clarify in the text
73. L558 the genus can be abbreviated to P. (also L731)
74. L560 to cure a plasmid
75. L565 how was the spontaneous curing performed/identified?
76. L571 of the Kr…
77. L601 in which RCR is controlled by the RepB….
78. L605 plasmid derived from
79. L609 Please use “data not shown” or “unpublished data” (also L642)
80. L629 Although incompatibility is the likely reason, could it also be a RM system target that is preventing establishment of two plasmids? This could be shown by using two selections (one for each plasmid) and then releasing the dual selection.
81. L635 when such replicons were introduced into
82. L640 Also, a control for the experiment
83. L641 vs L645 variable hyphenation is used for low-copy or high copy (please use one style throughout)
84. L649 where are the transposase data shown?
85. L648 65% after 150
86. L673 Overall, the PCN of
87. L674 of the pMEGA
88. L685 vs L686 variable hyphenation of soft-agar (I would include it)
89. L687 The strain
90. Table 7 and 695 and throughout; various styles used for “p”
91. L699/L700 “in static conditions” can be removed as it is implied from L698/L699
92. L706 since the time points have variable inclusion of “h” I woul dremove all instances until after 144 h.
93. L713 in Kr
94. L724 Please check that BHI has been defined on first use
95. L733 may limit the
96. L735 revealed that
97. L746 across the genus (when present), with over
98. L748 Additionally, we speculate an
99. L754 megaplasmids’ role
100. L772 whereas complete silencing was achieved for the phhbO gene (please also check the gene name; it should not be capitalized—and usually only have four letters)
101. L785 differences might highlight the importance
102. L790 searches using
103. L792 to L795 The authors might alert the readers that they are switching from talking about an acrR regulator (L792) to acr efflux pumps (L295).
104. L796, L797, L814, L815. These genera should be written in full on first use
105. L796 Typhimurium should not be italicized as it is a serovar not a species
106. L799 Please add a reference or link to this resource.
107. L807 are of broad interest
108. L818 In conclusion, the set up
109. L820 of the physiological roles of
110. L814 tools to shed light on
111. Some refs have full journal titles e.g. Ref 1, 8, 11, 16, 25, 31, whereas most others are abbreviated/truncated when there is more than one word. Please check throughout.
Comments on the Quality of English Language
Please see the examples above.
Author Response
Dear Reviewer,
Thank you very much for your thorough and constructive review of our manuscript. We are grateful for your positive assessment of our work, and we are pleased that you find our research of interest in streamlining Pseudoalteromonas haloplanktis TAC125 as versatile cell-factory. We appreciate your detailed feedback and suggestions that ensure that our manuscript is as clear and precise as possible for readers. We carefully addressed your valuable comments below, and your suggested adjustments are highlighted in red in the manuscript.
Comment 1: L22 and L26 I recommend using “of” rather than “from” e.g. L26 cured of the pMEGA plasmid
Response 1: Thank you, we adjusted the text accordingly.
Comment 2: L29 of the pMEGA
Response 2: Thank you, we adjusted the text as you suggested.
Comment 3: L31 to exploit valuable pMEGA genetic elements and further
Response 3: Thank you, we adjusted the text as “valuable pMEGA genetic elements” instead of “pMEGA valuable genetic elements”
Comment 4: L43 and L58 appear top conflict. The temp range is given as 20-30 in L43 but then 15 degrees L58. Please clarify in the text.
Response 4: Thank you for poiting out this mistake. We correctly reported the range of temperature as 4-20 °C as reported in the reference.
Comment 5: L45, L46, L94 Gamma (capitalized and italicized); please check throughout ms
Response 5: Thank you, we adjusted the format throughout the manuscript.
Comment 6: L61 a) a source
Response 6: Thank you, we adjusted in the text.
Comment 7: L62 [13,14], b)
Response 7: Thank you, the comma was added in the text.
Comment 8: L62 and c) an exceptional
Response 8: Thank you, the “as” word was removed as you suggested.
Comment 9: L73 chromosomal gene expression (it may seem counterintuitive since more than one gene would be expressed, but this is the standard style of referring to such “plurals”, for greater fluency).
Response 9: Thank you for your consideration. The plural was removed and the term “gene expression” was used.
Comment 10: L79 bacteria, including Pseudoalteromonas
Response 10: Thank you, the text was adjusted accordingly.
Comment 11: L86 vs L127 variable capitalization (and hyphenation) of rolling circle replication
Response 11: Thank you, “rolling-circle replication” was adjusted in the text.
Comment 12: L87 pMEGA encodes two type II
Response 12: Thank you, the verb “encodes” was changed in the text.
Comment 13: L88 YefM
Response 13: Thank you, the capital YefM was adjusted in the text.
Comment 14: L134 vs L136 different styles are used for “R” referring to resistance
Response 14: Thank you, done.
Comment 15: L139 experiment. The Ph…
Response 15: The article “The” was added in the text.
Comment 16: L140 was this mutant made in current study or was it already available. Please clarify.
Response 16: Thank you. We specified that it was already available.
Comment 17: L150 and throughout. I would write this as “Schatz salts” as written L701.
Response 17: Thank you for your suggestions, the term Schatz salts was substituted throughout the manuscript at L148, L223, L226, L244, L248, L250, L252, L270, and L343.
Comment 18: L156 through the heat-shock
Response 18: The article “the” was added in the text.
Comment 19: L171-2 Please define NCBI and NIH on first use; also CDS on L180, L199 MCS
Response 19: Thank you for pointing out the clarification. The following terms were added to the text accordingly: “National Institutes of Health”, “National Center for Biotechnology Information”, “coding sequence” and “multi cloning site”
Comment 20: L184 Please provide a reference for gateway cloning procedures used
Response 20: Thank you. The following reference was used “Hartley, J.L.; Temple, G.F.; Brasch, M.A. DNA Cloning Using In Vitro Site-Specific Recombination. Genome Research 2000, 10, 1788–1795, doi:10.1101/gr.143000.”
Comment 21: L186 Which PCR enzyme? Phusion (as listed L206)?
Response 21: Yes, the PCR enzyme used was Phusion High-Fidelity DNA polymerase form Thermo Fischer Scientific. The information was added to text.
Comment 22: L187 a pair of forward and reverse oligonucleotides listed (also L207)
Response 22: Thank you, “forward and reverse oligonucleotides” was adjusted in the text.
Comment 23: L206 and throughout; please provide company location on first mention. Please also include source of primers.
Response 23. Thank you for your suggestions. The source of primers (Eurofins Genomics) was included in L188, L208, and L298.
Comment 24: L211 used in this work
Response 24: Thank you, done.
Comment 25: L217 and throughout 100-mL Erlenmeyer
Response 25: Thank you for the suggestion. The 100-mL format was adjusted at L217, L224, L227, L248, L250, and L344.
Comment 26: L220 agar plates
Response 26: Thank you for your suggestion. The term “agar plates” was used instead of “agar plate”.
Comment 27: L222 25-mL inoculum
Response 27: Thank you, done.
Comment 28: L225 please define OD and provide wavelength on first mention
Response 28: Thank you, done.
Comment 29: L229 Please make sure IPTG is defined on first use
Response 29: Thank you. The IPTG was defined as Isopropyl β-d-1-thiogalactopyranoside in L229.
Comment 30: L229 and throughout. The authors use “hours” (in full) and then “h” either with a space or without. Please use one style throughout. I recommend using 8 h (for example)
Response 30: Thank you, done throughout the manuscript.
Comment 31: L231 and seeded on
Response 31: Thank you, done throughout the manuscript.
Comment 32: L233 replicated on a 70-mL
Response 32: Thank you, done.
Comment 33: L237 clones were subjected
Response 33: Thank you, done.
Comment 34: L239 for the PCR using
Response 34: Thank you, done.
Comment 35: L242 Growth Culture Conditions.. (please see note 9 above)
Response 35: Thank you, done.
Comment 36: L244 on a TYP
Response 36: Thank. Adjusted in the text.
Comment 37: L246 in 25-mL
Response 37: Thank you, done.
Comment 38: L248 in a 100-mL (also L250, L344); L252 in a 250-mL
Response 38: Thank you, done.
Comment 39: L256 culture set up
Response 39: Thank you, done.
Comment 40: L257 before inoculation
Response 40: Thank you, done.
Comment 41: L260 duplicates
Response 41: Thank you, done.
Comment 42: L262 vs section 2.10 Three different styles are used for writing the t-test. Please use one.
Response 42: Thank you, adjusted accordingly.
Comment 43: L287 plasmid intergeneric (please see note 9)
Response 43: Thank you, done.
Comment 44: L294, L302, L304, L307, L311 Company locations can be removed as given earlier
Response 44: Thank you, done.
Comment 45: L296 ampR (no capitalization)
Response 45: Thank you, done.
Comment 46: L297 Each pair of primers
Response 46: Thank you, done.
Comment 47: L298 Please give a reference or source of Primer 3
Response 47: Thank you, done.
Comment 48: L298 Table S1)
Response 48: Thank you, done.
Comment 49: L306 10-uL mixtures
Response 49: Thank you, done.
Comment 50: L329 the zone of inhibition was measured in cm. The experiment was
Response 50: Thank you for your helpful suggestions regarding the phrasing of “disk of inhibition.”
Looking for a proper term into literature, we found that both "area of inhibition" and "zone of inhibition" are widely used terms in microbiological studies. After considering both options and ensuring consistency with terminology commonly found overall in microbiology literature, we have opted to use "area of inhibition." We hope this decision addresses your concerns, and we appreciate your understanding.
Comment 51: L336 10 min
Response 51: Thank you, done.
Comment 52: L362 has “wt” been defined?
Response 52: Thank you, I defined wild type for the first time in L362.
Comment 53: L383 184,073-bp long
Response 53: Thank you, done.
Comment 54: L392 to the parA
Response 54: Thank you, done.
Comment 55: Table 3. It is unclear why the hsd proteins were not identified by BlastP if they were found by Blastn
Response 55. Thank you for pointing out that this information was missing. The hsd protein was indeed identified, but this required a manual approach. In fact, when using the automated BlastP system, the protein was not detected, likely due to limitations in the annotation process or inconsistencies in the NCBI database annotations. Instead, when individual hsd protein sequences (from pMEGA and pPsKG3-1) were subjected to a targeted BlastP search, we were able to successfully identify it as “unnamed protein product” which results to be annotated as “Type I restriction endonuclease subunit R” in the FASTA file of the pPsKG3-1 proteins. We hope this explanation clarifies the approach we used, and we are happy to further elaborate if needed. The information was added to the table 3 in the new version of the manuscript.
Comment 56: L413 encoded genes
Response 56: Thank you, done.
Comment 57: L475 with multiple genes
Response 57: Thank you, done.
Comment 58: L484 259-bp
Response 58: Thank you, done.
Comment 59: L486, L498 vs L827 Three different styles used for eGFP
Response 59: Thank you, this was adjusted throughout the manuscript.
Comment 60: L469 RCR
Response 60: Thank you, done.
Comment 61: L486, L498 different styles are used for placZ (I would use the style L486)—may have been an earlier example of PlacZ (please check)
Response 61: Thank you, it was adjusted throughout the manuscript.
Comment 62: L490 I would combine these 2 paragraphs
Response 62: Done. Thank you for your suggestion.
Comment 63: L503 inserts into the umuC (please check italicization of “C” in L504 also)
Response 63: Thank you, done.
Comment 64: L509 selection of mutants by this approach, not feasible.
Response 64: Done. Thank you for your suggestion.
Comment 65: L509 vs L501 CamR or cmR
Response 65: Thank you, it was adjusted throughout the manuscript.
Comment 66: L514 different pairs
Response 66: Thank you, done.
Comment 67: L517 vector inserted
Response 67: Thank you, done.
Comment 68: L519, L521, L539 three different styles are used for writing antisense RNA. Please unify throughout ms.
Response 68: Thank you. The term “antisense” was adjusted throughout the manuscript.
Comment 69: L521 Please define RBS on first use
Response 69: Thank you, done.
Comment 70: L525 against the
Response 70: Thank you, done.
Comment 71: L547 in the pMEGA
Response 71: Thank you, done.
Comment 72: L554 what is “clean PCR”? please clarify in the text
Response 72: Thank you for your suggestion. It was specified that a purified template was used.
Comment 73: L558 the genus can be abbreviated to P. (also L731)
Response 73: Thank you, done.
Comment 74: L560 to cure a plasmid
Response 74: Thank you, done.
Comment 75: L565 how was the spontaneous curing performed/identified?
Response 75: Thank you. The following statement was added to the text “selecting on TYP agar in presence and in absence of ampicillin antibiotic.”
Comment 76: L571 of the Kr…
Response 76: Thank you, done.
Comment 77: L601 in which RCR is controlled by the RepB….
Response 77: Thank you, done.
Comment 78: L605 plasmid derived from
Response 78: Thank you, done.
Comment 79: L609 Please use “data not shown” or “unpublished data” (also L642)
Response 79: Done. Thank you for your suggestion. The term unpublished data was used.
Comment 80: L629 Although incompatibility is the likely reason, could it also be a RM system target that is preventing establishment of two plasmids? This could be shown by using two selections (one for each plasmid) and then releasing the dual selection.
Response 80: Thank you for your valuable observation. Upon reviewing the text, I recognize that our phrasing might have unintentionally led to confusion regarding the purpose of this experiment. While we make an improper use of the term "incompatibility," the primary focus of this analysis is to explore the applicability and replicability of rep-based plasmids and evaluate their behavior examined strains. I have rephrased the paragraph to reflect the actual intent of the experiment. I hope that this rephrasing makes the intention clearer and better conveys the goal of investigation of rep-based plasmids in the context of the pMEGA replication machinery. You can find the adjustment at L599-L640.
Comment 81: L635 when such replicons were introduced into
Response 81: Thank you, done.
Comment 82: L640 Also, a control for the experiment
Response 82: Thank you, done.
Comment 83: L641 vs L645 variable hyphenation is used for low-copy or high copy (please use one style throughout)
Response 83: Thank you, done.
Comment 84: L649 where are the transposase data shown?
Response 84: (lines 644-648) Thank you for pointing out the confusing statement. We acknowledge that we were not entirely clear in the original text, and we have rephrased the paragraph to improve clarity on why we expect pMEGA transposase, possibly along with other genes, to be involved in heterologous plasmid stability. See L644-648.
Comment 85: L648 65% after 150
Response 85: Thank you, done.
Comment 86: L673 Overall, the PCN of
Response 86: Thank you, done.
Comment 87: L674 of the pMEGA
Response 87: Thank you, done.
Comment 88: L685 vs L686 variable hyphenation of soft-agar (I would include it)
Response 88: Done, thank you. The hyphenation of soft-agar was included.
Comment 89: L687 The strain
Response 89: Done, thank you.
Comment 90: Table 7 and 695 and throughout; various styles used for “p”
Response 90: Done, thank you.
Comment 91: L699/L700 “in static conditions” can be removed as it is implied from L698/L699
Response 91: Thank you, done.
Comment 92: L706 since the time points have variable inclusion of “h” I would remove all instances until after 144 h.
Response 92: Thank you, done.
Comment 93: L713 in Kr
Response 93: Done, thank you.
Comment 94: L724 Please check that BHI has been defined on first use
Response 94: Sorry, this term was a typo and changed with TYP.
Comment 95: L733 may limit the
Response 95: Done, thank you.
Comment 96. L735 revealed that
Response 96: Done, thank you.
Comment 97: L746 across the genus (when present), with over
Response 97: Done, thank you.
Comment 98: L748 Additionally, we speculate an
Response 98: Done, thank you.
Comment 99: L754 megaplasmids’ role
Response 99: Done, thank you.
Comment 100: L772 whereas complete silencing was achieved for the phhbO gene (please also check the gene name; it should not be capitalized—and usually only have four letters).
Response 100: Thank you, L772 was rephrased as suggested. The cited hbo gene is a P. haloplanktis TAC125 gene and the Ph before hbo stands for Pseudoalteromonas haloplanktis. We used the same style as reported in literature.
Comment 101: L785 differences might highlight the importance
Response 101: Thank you, done.
Comment 102: L790 searches using
Response 102: Done, thank you.
Comment 103: L792 to L795 The authors might alert the readers that they are switching from talking about an acrR regulator (L792) to acr efflux pumps (L295).
Response 103: Thank you very much for your suggestion. Your feedback helped me recognize that the paragraph may have come across as unclear and I might have missed the point. To address this, I have rephrased the paragraph to streamline the content and make it more clear while maintaining its intended meaning.
Comment 104: L796, L797, L814, L815. These genera should be written in full on first use
Response 104: Done, thank you.
Comment 105: L796 Typhimurium should not be italicized as it is a serovar not a species
Response 105: Done, thank you.
Comment 106: L799 Please add a reference or link to this resource.
Response 106: Thank you, done.
Comment 107: L807 are of broad interest
Response 107: Done, thank you.
Comment 108: L818 In conclusion, the set up
Response 108: Thank you, done.
Comment 109: L820 of the physiological roles of
Response 109: Done, thank you.
Comment 110: L814 tools to shed light on
Response 110: Thank you, done.
Comment 111: Some refs have full journal titles e.g. Ref 1, 8, 11, 16, 25, 31, whereas most others are abbreviated/truncated when there is more than one word. Please check throughout.
Response 111: Thank you for pointing out this variability. We check throughout the references list and uniformed the style.
Reviewer 4 Report
Comments and Suggestions for Authors
The manuscript «Engineering the Marine Pseudoalteromonas haloplanktis TAC125 via pMEGA Plasmid Targeted Curing Using PTasRNA Technology» submitted by Angelica Severino et al. to Microorganisms is devoted to the investigation of the role of pMEGA plasmid and the future
biotechnological applications of PhTAC125 in recombinant protein production. Pseudoalteromonas haloplanktis TAC125 (PhTAC125) is marine bacterium adapted to thrive in extreme environments. This bacterium has a unique biotechnological potential.
Introduction is devoted to description of Pseudoalteromonas haloplanktis
TAC125 (PhTAC125) as the most valuable marine microorganisms, and its possible usage as
a model organism of cold-adapted bacteria.
The section Materials and methods contains all necessary protocols. This part is written with technical details those are important for the experiments.
The section Results is very good illustrated and contains full description of experiments and their results.
Discussion is in accordance with the obtained results and conclusions are confirmed by the experimental results.
Authors successfully removed the pMEGA plasmid from the bacterium and obtained a novel strain named KrPL2. This study opens up the possibility for the construction of new genetic tools exploited of microorganisms from polar ecosystems for biotechnological purposes.
I have some questions and remarks:
1) The manuscript does not contain the section conclusions. I think it will be interesting to summarized the results and write some future perspectives for their possible applications.
2) There are some problems with the tables. Some numerous are placed above the text in first column in the table (Table 4,6 and 7).
Author Response
Dear reviewer,
I sincerely thank you for your thoughtful and thoughtful review of our manuscript. We are truly grateful that you consider our study relevant, well-written, and informative. We truly appreciate your positive assessment of our work and your recognition of its significance in advancing biotechnological applications of Pseudoalteromonas haloplanktis TAC125.
We acknowledge your questions and remarks, and we have addressed them in the manuscript. You’ll find our changes highlighted in blue.
Comment 1: The manuscript does not contain the section conclusions. I think it will be interesting to summarized the results and write some future perspectives for their possible applications.
Response 1: (lines 842-856) Thank you for your precious suggestion. The following paragraph named “Concluding remarks” was added to the manuscript at lines 842-856:
“5. Concluding Remarks
In this study, we developed a novel strain of P. haloplanktis TAC125, named KrPL2, by successfully curing the megaplasmid pMEGA through a dual genetic approach combining homologous recombination and PTasRNA technology. The cured strain exhibited enhanced resistance to oxidative stress, which is a critical feature for biotechnological applications, especially in industrial fermentation systems where oxidative stress is a limiting factor. Additionally, the reduced capability of biofilm formation offers potential advantages in industrial settings by improving process efficiency and reducing contamination risks. Finally, the streamlined genomic background of KrPL2 provides a robust platform for designing and testing new psychrophilic genetic tools leveraging the unique properties of psychrophilic bacteria genetic features. In conclusion, the set-up of a robust curing strategy may represent a significant achievement for the understanding of the physiological role of megaplasmids in Antarctic bacteria and highlights potential biotechnological applications of PhTAC125.”
Comment 2: There are some problems with the tables. Some numerous are placed above the text in first column in the table (Table 4,6 and 7).
Response 2: Thank you for pointing out the issue. The table placing was fixed.
Reviewer 5 Report
Comments and Suggestions for Authors
This manuscript [“Engineering the Marine Pseudoalteromonas haloplanktis TAC125 via pMEGA Plasmid Targeted Curing Using PTasRNA Technology”] by Severino et al. intends to streamline the host genetic background by curing Pseudoalteromonas haloplanktis TAC125 (PhTAC125) from the endogenous megaplasmid (pMEGA) using a sequential genetic approach. The authors combined homologous recombination by exploiting a suicide vector with the PTasRNA gene silencing technology to interfere with pMEGA replication machinery. In short, the present study is relevant in terms of its future biotechnological potential, the manuscript is well-written and as a whole quite informative as well. I, therefore, recommend it for publication in the journal of Microorganisms.
Author Response
We are in debt with the reviewer for the time she/he spent reading our manuscript. Thanks for the comments.
Round 2
Reviewer 1 Report
Comments and Suggestions for Authors
I consider that the authors improve their manuscript and it can be accepted in the present form.